# Random fractal-enabled physical unclonable functions with dynamic AI authentication

Ningfei Sun [1,2], Ziyu Chen [1], Yanke Wang [3], Shu Wang[2], Yong Xie [1,4] ✉ & Qian Liu [2] ✉

A physical unclonable function (PUF) is a foundation of anti-counterfeiting processes due to its inherent uniqueness. However, the self-limitation of conventional graphical/spectral PUFs in materials often makes it difficult to have both high code flexibility and high environmental stability in practice. In this study, we propose a universal, fractal-guided film annealing strategy to realize the random Au network-based PUFs that can be designed on demand in complexity, enabling the tags' intrinsic uniqueness and stability. A dynamic deep learning-based authentication system with an expandable database is built to identify and trace the PUFs, achieving an efficient and reliable authentication with 0% "false positives". Based on the roughening-enabled plasmonic network platform, Raman-based chemical encoding is conceptionally demonstrated, showing the potential for improvements in security. The configurable tags in mass production can serve as competitive PUF carriers for high-level anti-counterfeiting and data encryption.

Anti-fake labels as the authentication tools of product authenticity face an increasing challenge in security, and researchers are developing new secure anti-counterfeiting methods, continuously pushing the anti-fake science and technology ahead. However, forgery and counterfeiting still result in great losses worldwide, damaging the normal order of the market or even human safety[1,2]. For instance, counterfeit electronics involving network security, counterfeit pharmaceuticals, and so forth can cause an estimated trillion-dollar in losses each year[1]. One major source of these losses is that current labels are often subject to being counterfeited due to their deterministic fabrication mode[3,4]. Thus, developing unclonable security tags based on a new principle would be fundamentally important against the counterfeiting of labels.

Physical unclonable functions (PUFs; i.e., physical one-way functions)[1,5,6] have become promising identifiers for high-fidelity labels and digital storage. A PUF refers to a physical object with inherent, unique, and fingerprint-like features that are generated via a stochastic and non-deterministic process[1]. The intrinsic randomness ensures sufficient complexity and a high encoding capacity of PUFs, making them nearly impossible to be duplicated. To date, various types of PUFs have been developed, including (i) directly visualized graphical PUFs composed of randomly distributed micro/nanostructures (e.g., wrinkling[7–10]/buckling[11]/folding[12]-based artificial fingerprints, randomly formed evaporative patterns[13–15], and randomly arranged micro/nanoparticles[16–20]); (ii) spectral PUFs with the aid of an analytical tool for readout (e.g., random stimulated luminescence[21,22], surface-enhanced Raman scattering (SERS) patterns[23,24], irregular texture[25]/matrix[5,26] linear scattering-based speckle patterns, and chaotic nonlinear silicon photonic devices[27,28]); and (iii) complex electronic PUFs with diverse disorders and inherent imperfections (e.g., graphene[29] or randomly distributed carbon nanotube-based[30] field-effect transistors, oxide or halide-based memristors with intrinsic entropy sources[6,31,32]). Among these PUFs, graphically encoded tags primarily focused on surface information are more convenient and robust in identification due to direct imaging by simple optical microscopy. Particularly, the levels of the PUF tags in complexity can be actively regulated only by recording different pattern areas or changing the physical feature size[7–9,19]. For example, a random wrinkle system has flexible controllability in code complexity by modulating

[1]School of Physics, Beihang University, Beijing 100191, China. [2]CAS Center for Excellence in Nanoscience, National Center for Nanoscience and Technology & University of Chinese Academy of Sciences, Beijing 100190, China. [3]Institute for Automation and Applied Informatics, Karlsruhe Institute of Technology, Karlsruhe 76344, Germany. [4]Key Laboratory of Intelligent Systems and Equipment Electromagnetic Environment Effect, School of Electronic and Information Engineering, Beihang University, Beijing 100191, China. ✉e-mail: xiey@buaa.edu.cn; liuq@nanoctr.cn

wrinkle instability[7,9,33], and thus, it has a configurable encoding capacity that can be sufficiently adaptable to an on-demand encryption strategy. However, the surface textures of these flexible material-based tags often have relatively low physical robustness in practice, such as elastomeric polymer physical aging at high temperature, humidity/water, or oxygen[9,34]. Therefore, ideal graphical PUF carriers compatible with both high environmental stability and flexibility remain to be developed.

Random fractal structures widely exist in various irregular morphologies in nature, owning highly random topography and statistical self-similarity between the local and whole geometries[35–37]. The fractal theory is also used to elucidate the complex surface morphology evolution of diverse thin film systems, such as fractal-guided percolation networks/clusters of films[36–38]. A Percolation network/cluster refers to a system in global connectivity through a continuous "chain" of locally connected objects, such as self-assembled Au nanoframeworks[36], Au clusters through film deposition[37], and film annealed-induced Au islands[39]. Thermal annealing of the gold film can induce the sequential surface morphology evolution from film rupturing to spontaneously shrinking into randomly ramified structures below the percolation threshold[38,39]. The ramified percolation structures possess typical fractal traits such as intrinsic randomness and unpredictability, implying their qualifications as inimitable PUF tags. More importantly, the stable physicochemical properties of gold ensure the durability of the labels under extreme conditions.

On the other hand, tags with multiple responses can have a higher security level[1,12,17–20]. Combining chemistry with the PUFs is an ideal strategy to achieve this goal by mixing various types of taggants (e.g., stimuli-responsive molecules) with multiple detectable chemical characteristics[1,21,24,40]. Plasmonic nanostructures can be used as effective carriers for further increasing the security level since they have stable localized surface plasmon resonance (LSPR), which can generate an enormous enhancement of the electromagnetic field under light excitation and amplify the chemical signals[41], such as randomly arranged Raman-probe-embedded plasmonic nanoparticles[17,24]. Yet the unpredictable signal extraction sites and particle aggregations pose a challenge to efficient authentication[42,43]. Plasmonic nanoparticles can also be incorporated into the graphically encoded hosts, such as randomly folded[12] or de-wetted polymer systems[20], endowing the polymer hosts with orthogonal chemical spectral information. However, nanoparticles usually have indirect chemical-assisted bonding with the object, causing low stability under external conditions[44]. Nonuniform particle distribution on the object surface can also lead to a loss of information[20], which affects the robustness of the readout. Therefore, an inherent and homogeneous plasmonic platform independent of chemically synthesized nanoparticles is desirable.

Also, constructing an efficient and reliable authentication system for identifying the security key is essential in anti-counterfeiting. Automated image identification is appropriate to decode the PUF keys. Conventional image processing algorithms are based on pattern recognition and comparison analysis[1]. Apart from the relatively tedious matching time, their performance also strongly depends on the image orientation and quality[44,45]. Deep learning (DL)[11,13,18], as an artificial intelligence (AI) technique, has been popularly used to validate the security key through trained neural networks with high authentication efficiency and accuracy as well as high readout toleration under different conditions. The PUF system requires a record of all the PUF keys. However, it is time-consuming to train the deep learning model with a large PUF key database. Therefore, continuous improvement is required in the back-end exploitation of the existing deep learning-based PUF authentication systems.

In this study, we develop an efficient anti-counterfeiting system based on the random fractal-network PUFs and an AI authentication used for authenticity identification. Combined with laser lithography, multiple network tags can be simultaneously integrated on the substrate through one-step annealing of the Au film. These Au networks can be flexibly configured in terms of wavelengths and amplitudes by changing the film thickness, which allows us to design the structural complexity on demand. An effective encoding capacity of $10^{348}$ is realized, and the capacity value can be raised by recording a larger pattern area and denser network feature. The surface of the Au network is roughened at the nanoscale and thus can generate enhanced electromagnetic "hotspots" under light excitation. The proof-of-concept of SERS-based chemical encoding shows the feasibility of the Au-based PUF in multiple-level encryption. Finally, a convenient (smartphone readout), fast (authentication in 6.36 s), and reliable (zero "false positives" case) deep learning-based authentication system is presented to identify and trace the PUF tags. We also propose a dynamic key database strategy to simplify the tedious training procedure of the deep learning model, which has a lot of potentials to manage a larger PUF key database. Thus, the comprehensive PUF labels that comply with the demands of inherent uniqueness, code reconfigurability, multiple-level security, mass production, and environmental stability (e.g., tolerable in extreme temperature, water/humidity, and abrasion), along with efficient and reliable AI authentication system, will pave a broad avenue toward the applications in next-generation anti-counterfeiting.

## Results
### Random fractal-enabled PUFs
The generation of the random fractal-guided Au network structure is described schematically in Fig. 1a. Using laser direct writing (LDW) and magnetron sputtering, we can realize the Au film into arbitrary shapes with a defined thickness on a Si/SiO₂ substrate and then follow a thermal annealing process to obtain the randomly arranged Au networks as the PUFs (Fig. 1a). Representative bright- and dark-field optical images show the unique fractal-like network tag (Fig. 1b, c). Importantly, as an effective measure of protecting the physical object from damage in practice, the PUF tag can be covered with a transparent thin film according to the usage environment (Fig. 1a), such as SiO₂, Al₂O₃, and poly(methyl methacrylate) (PMMA).

The fractal-guided surface depercolation process of the Au film as a function of annealing time is shown in Supplementary Fig. 1. Randomly distributed voids are first formed due to the strain instabilities as a result of the thermal expansion mismatch between the Au film and the underlying support[46,47]. Then, the void edges in different orientations stochastically retract toward an equilibrium state below the percolation threshold via edge curvature-induced tension gradients until they stop at the bifurcations successively[38,48]. Irregular and complex mesh-like structures are then formed (Supplementary Fig. 2), which involve multiple randomly ramified percolation clusters with different correlation lengths and typical fractal characteristics[37,39]. Some nearly spheroidized Au particles also exist due to the local heterogeneities of the film. X-ray diffraction patterns (Supplementary Fig. 3) reveal the evolution of crystallinity from the polycrystalline structure of the Au film to the approximately single-crystalline structure of the Au network, which finally tends to become spherical Au particles[38].

The branching fractal model is exhibited in Fig. 1d. A single PUF tag is composed of distinct fractal objects, and several iterated bifurcations make up one fractal. Upon each iteration, two new branches are added to the terminal branch. The extension directions and lengths of the branches are randomly varied but exhibit statistical feature similarity in different bifurcations. For example, in Fig. 1d, the bifurcations of part 1 and the derived part 2 have a similar quasi "Y-shape" feature. Such a fractal-guided depercolation process ensures the impossibility of duplicating the random network structures, making these "fingerprints" immune to attacks. Even though, film thickness-dependent annealing temperature and corresponding annealing time can ensure the high reliability (a measure of reproducibility) of the

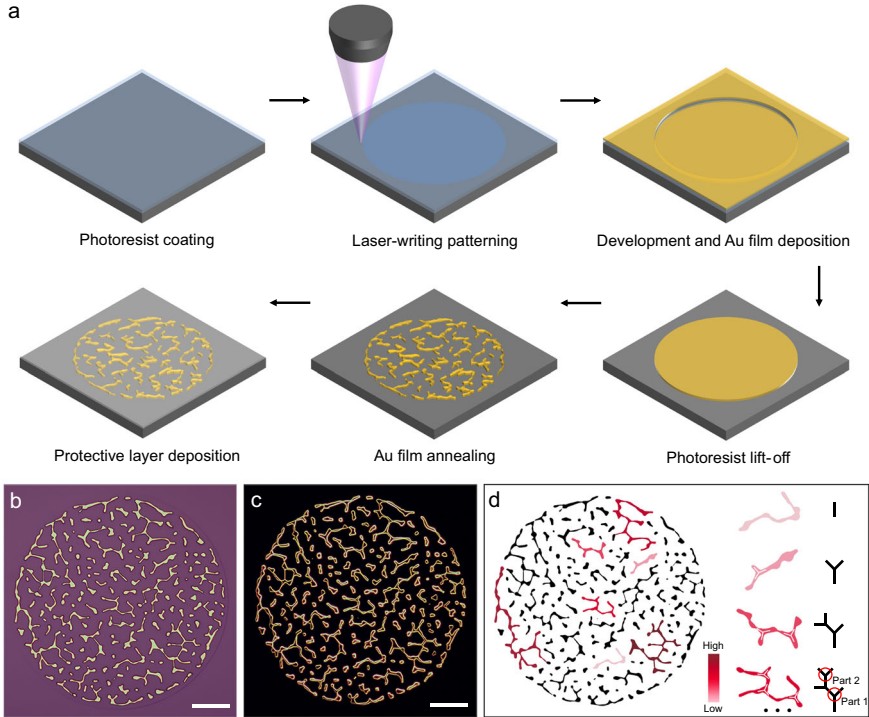

**Fig. 1 | Fabrication and characterization of random fractal-guided PUFs. a** Flow chart of the PUF tag fabrication. The Si/SiO$_2$ substrate was first spin-coated with one layer of photoresist (step 1), followed by laser-writing the circular films as basic units (step 2). After the development of the patterning area, one layer of Au film was then deposited on the substrate through magnetron sputtering (step 3), followed by the photoresist lift-off procedure (step 4). Next, the Au film in the circular area was annealed to obtain the random fractal-like Au networks (step 5). Finally, one layer of the optically transparent thin film was spin-coated on the tag as a protective layer (step 6). **b** Optical micrograph of the annealing-induced random Au network tag as the PUF. **c** Dark-field micrograph of **b**, which shows a stronger visual contrast of the structure profile. **d** Cayley tree-like fractal structure analysis. As the degree of redness deepens, the fractal order increases gradually. The distinct fractals are extracted from the binarized network pattern. The corresponding mathematical models from orders 0 to 3 are also presented as a comparison. Scale bars: 20 μm (**b**, **c**).

fabricated PUF tags with the same production parameters (Supplementary Fig. 4).

## Configurable encoding design

Similar to the elastic wrinkle system[7,8], these random rigid network structures can also be flexibly configurable in terms of amplitude and wavelength by changing the film thickness and corresponding annealing parameters. Thermal fluctuations induced by the interface interactions between the film and the substrate can account for the spontaneous rupture of the original Au film[47,49]. The induced perturbative oscillation across the surface that causes the deformation of the film can be quantitatively represented by the critical wavelength $\lambda c$[47],

$$\lambda c = h_0^2 \sqrt{\frac{4\pi^3\sigma}{A}} \tag{1}$$

where $h_0$ is the thickness of the film, $\sigma$ is the temperature-dependent surface tension of the film, and $A$ is the Hamaker constant of the film on the specific support. The film fragment with a span smaller than the critical wavelength tends to be stable. According to this physical model, as the thickness increases, the critical wavelength lengthens (i.e., the ruptured Au-film fragment can have a larger size to maintain a state of thermal stability)[49]. Therefore, the sparser network with a wider wavelength can be contracted from the thicker Au film, which is consistent with the experimental results shown in Fig. 2a–d. A typical 3D topographical image of the Au network pattern (Fig. 2e) and the corresponding cross-sectional profile (Fig. 2f) shows the relatively uniform network wavelength and amplitude. The height information of the network can also be encoded into the PUFs and decoded via structure height characterization. Figure 2g shows an increase in the

amplitude and wavelength of the network with an increase in film thickness, and these parameters might be anticipated to be extended with broader variation ranges.

In addition to the visual contrast of the pattern complexity, the fractal dimension is widely used to quantitatively characterize the complexity of fractal thin film structures[36,50]. In the PUF patterns, there exists an increase in fractal dimension ($D$; see details in Supplementary Fig. 5) with a decrease in film thickness from 1.52 at 90 nm to 1.75 at 30 nm. Therefore, the complexity of the network tag can be designed on-demand by regulating the film thickness. In this study, four typical thicknesses were implemented to exhibit the configurability in code complexity, and theoretically, the thickness can be further subdivided to systematically classify the security level (e.g., low, medium, or high level) based on the network wavelength for different applications. The size of the PUF tags can also be flexibly regulated to control the security level (Supplementary Fig. 6).

Also, random networks can be generated in various geometries via a deterministic production mode. Laser direct writing can satisfy the unlimited flexibility in shape design, such as square, triangle, pentagon, and pentagram (Fig. 2h). Therefore, it can be additionally used in categorically labeling an abundant variety of goods or designing customized trademarks. Collectively, the high controllability of the network-based PUF tag in internal physical features and external geometric configurations is demonstrated, allowing us to design the security levels and classify a randomly generated anti-counterfeiting tag on demand.

## Performance of the PUFs

We calculated the cross-correlation values of 600 patterns from the same batch of production using the feature similarity algorithm

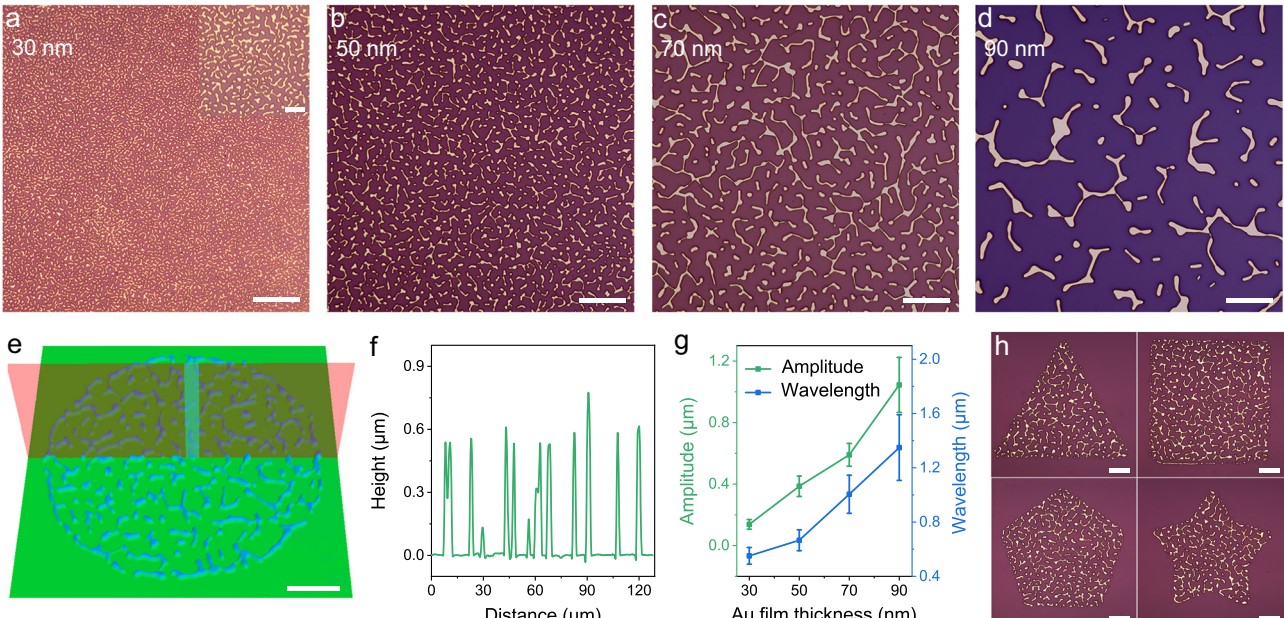

**Fig. 2 | Configurable network topography. a–d** Optical micrographs of the random Au networks with different film thicknesses (from 30 to 90 nm). **e** Typical 3D topography of an isolated PUF tag for the 70-nm-thick Au film and **f** cross-sectional profile along the diameter. **g** Amplitude and wavelength of the networks with different Au film thicknesses, both showing a gradually increasing tendency with an increase in film thickness. The error bars represent the standard deviations of the independent data, which correspond to the small fluctuation of the network size. **h** PUF patterns with various geometries for classification. Scale bars: 5 μm (**a** inset) and 20 μm (**a–d**, **e**, **h**).

(FSIM)[51] to verify the uniqueness of the PUF tag. The cross-correlation map shows a typical diagonal feature (Fig. 3a), which reveals that a high topographical feature correlation value (i.e., the high intracorrelation values and the rest data are expressed as the intercorrelation values) only comes from the same pattern. Cross-correlation values of patterns from different batches under identical fabrication parameters were also evaluated to verify the inherent uniqueness along the consecutive production process (Supplementary Fig. 7). The distribution histogram of the cross-correlation values in Fig. 3b also shows a clear separation between the intracorrelation and intercorrelation values, implying that the random network tags are particularly suitable for unclonable anti-counterfeiting labels due to their unique topographical features and the unlimited number of different topographies. The 3-dimensional feature information can be encoded into every pixel of the PUF pattern, while each pixel is used as a variable in different grayscale intensities derived from the structure height-dependent different light reflection and refraction. However, due to the inevitable limitation of the imaging unit in image contrast or color resolution, the physical features in practically captured PUF images usually cannot possess a theoretical grayscale distribution from 0 to 255. To present a more general level of grayscale encoded in physical features, we calculated the grayscale histograms from 10 randomly selected PUF images in the basic database, subtracted the grayscale information that the feature structure cannot reach, and counted an average value of 140 as the valid level of grayscale in a practical PUF image (valid grayscale distribution from 74 to 214, as shown in Supplementary Fig. 8). By converting an illustrational image to a size of 750 × 750 pixels, an effective encoding capacity of $10^{348}$ was estimated according to Carro–Temboury's universal model[21] (see calculation details in Supplementary Fig. 9 and Supplementary Note 1), which is much larger than a basic PUF encoding capacity of $10^{20}$ and has good anti-counterfeiting effectiveness[1].

The encoding capacity of the PUFs is not dominant compared to the wrinkling/crumpling system with dynamic grayscale information or randomly distributed stimuli-responsive taggants with multiple responses. However, a PUF with an extremely large encoding capacity can reduce the probability of fabricating two tags with the same configuration by a stochastic process (i.e., a low reproducibility) and generating the same response. The ultrahigh complexity and poor stability also cause difficulty in cryptography applications. We can also achieve a configurable encoding arrangement (Fig. 3c) due to the flexible design in the complexity of the networks. Within a fixed area of the image, the PUF pattern can be regulated in size (coverage area of networks) and Au film thickness (filling ratio of networks) to realize different capacities (Supplementary Fig. 10 and Supplementary Note 1), which is sufficiently suitable for a customized encryption strategy for different applications. The captured image size and contrast may be decreased by inevitable interference from the imaging unit, which can be alleviated by employing PUFs with denser feature information (Fig. 2a–d) or image preprocessing of denoising and grayscale stretch.

Interestingly, the random Au networks show unique features that can not only be graphically displayed via pixel-based microscope images but also be spectrally monitored through SERS response, highlighting the potential of realizing multiple-level security. As a typical noble metal, Au has strong LSPR properties at the nanoscale[41]. However, the micron-sized network structure introduced in this study, similar to bulk Au to some extent, exhibits a poor LSPR effect. Through a straightforward ion bombardment on the network surface by the ion cleaning technique (Fig. 3d), we can subtly fabricate dense and uniform convex Au nanostructures on the surface, which can be used as the "hot spots" of the electromagnetic field enhancement and regarded as a reliable SERS substrate. The 3D atomic force microscopy (AFM) image in Fig. 3e shows the roughened surface with a height fluctuation of approximately 1.5 nm. Scanning electron microscopy (SEM) images in Supplementary Fig. 11 show the apparent topography contrast before and after the roughening treatment. Finite-difference time-domain (FDTD) simulation (Supplementary Fig. 12) reveals the intrinsic electromagnetic field enhancement on the roughened surface under light excitation, theoretically predicting the effectiveness of this versatile plasmonic platform. The Raman response of rhodamine 6G (R6G) (Fig. 3f) and Raman spectra collected from the networks before

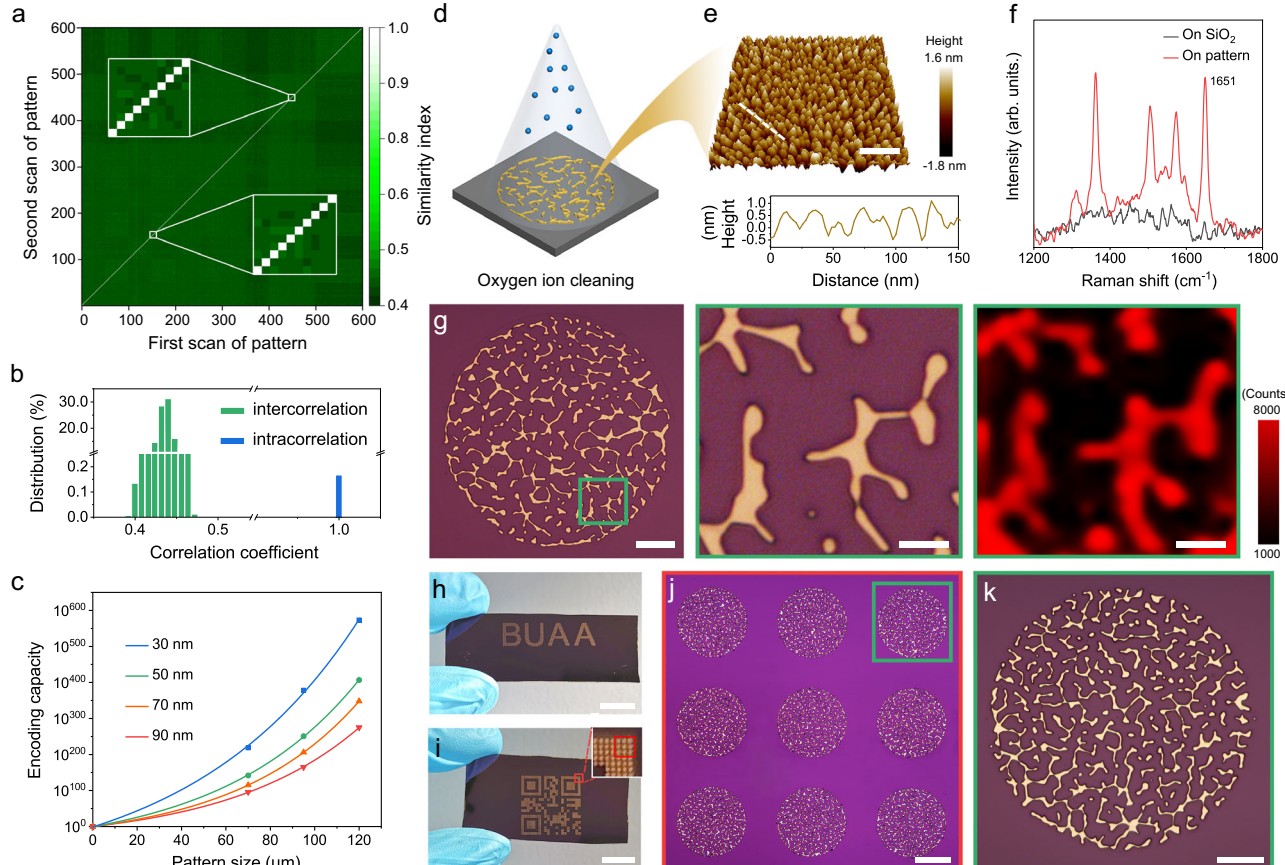

**Fig. 3 | Individuality characterization and multiple-level security strategy of the PUFs. a** Heat map showing the cross-correlation values obtained from 600 network patterns among the same fabrication batch. Each pattern was scanned twice, and calculations were carried out between these two image sets. The color bar represents the similarity index, the data along the diagonal line represent the values from the same images, and the others represent the values from different images. **b** Distribution of the correlation coefficients from (**a**). **c** Encoding capacity of the PUF tags with different film thicknesses and pattern sizes, showing exponential growth in terms of the pattern size, which is well fitted by the ExpDec 1 function. **d** Schematic illustration of the surface roughening process of the Au network via oxygen ion cleaning. **e** 3D AFM image of the surface morphology of the roughened Au network and the corresponding cross-sectional profile with a height fluctuation of approximately 1.5 nm. **f** Raman spectra of the R6G probe molecules from the Si/SiO₂ substrate and roughened Au network surface. **g** Raman mapping of the R6G molecules orthogonal to the physical feature. The color contrast remains consistent for quantification. Macroscopic graphics of **h** letters "BUAA" and **i** QR code composed of **j** isolated PUF patterns. **k** Enlarged image of one typical PUF pattern from **j**. Scale bars: 100 nm (**e**), 20 μm (**g** the first image, and **k**), 4 μm (**g** right two images), 1 cm (**h**, **i**), and 60 μm (**j**).

and after the surface treatment (Supplementary Fig. 13) verify that surface roughening stimulates the active LSPR of the pristine Au network. Raman mapping in Fig. 3g provides direct insight into the spatial distribution of the SERS response, demonstrating the robust readout of the chemical tag orthogonal to the physical feature of the plasmonic network. Thus, as a proof-of-concept, the SERS-based chemical information encoded into the hierarchical networks with unique identifiers can also be employed as an auxiliary security layer, enabling the plasmonic network to be a potential multidimensional PUF tag with a higher encoding capacity and non-replicability.

Also, the naked-eye visible tags integrated with the basic PUF units were manufactured by the mask-assisted lithography technique. Figure 3h, i display the customized patterns with the logo of Beihang University (BUAA) and the QR code (carrying additional information), respectively. While the macroscopic graphics are easy to authenticate, they are also simple to counterfeit. Nevertheless, at the microscopic level, in this case, all basic units are completely distinct from each other, showing a unique and unclonable physical feature (Fig. 3j, k).

In addition, commercial security labels should also meet several requirements, such as high environmental stability, mass production, and low cost[1]. With its intrinsic stability due to the relativistic contraction of the 6 s electron shell[52], Au has excellent antioxidation and chemical anti-corrosion properties. Considering the importance of the

label's stability at extreme temperatures (e.g., applications in cold chain logistics or high-temperature service)[53], we tested the temperature durability of the proposed PUF labels. The label was refrigerated at −40 °C in the atmosphere for 60 h, showing no variation in topographical features (Supplementary Fig. 14). To test the long-term thermal stability, the same PUF label was consecutively heated in the atmosphere from 600 to 775 °C for 10 h with an interval of 50 °C. The Au-based label remained stable up to 750 °C but began to change at 775 °C (Supplementary Fig. 15), which is marginally below the original label production temperature of 800 °C, showing superior stability at high temperatures. The security label should also survive inevitable ambient exposure in daily life[15,19], such as mechanical abrasion, aqueous corrosion, and contamination. All these can be simply implemented by covering a protective layer, such as SiO₂ or Al₂O₃ for high-temperature situations, PMMA for daily temperature situations, and others on the PUF tag according to the specific use cases. In this study, the label covered with PMMA was evaluated to demonstrate whether it has an excellent anti-scratch, waterproof, and anti-dust performance. The label underwent sonication in water and repetitive physical friction and showed no variation in physical features (Supplementary Fig. 16). Dust and stains generated from the real-world environment can also be cleanly removed by wiping with a soft cloth (Supplementary Fig. 17). In addition, the PMMA layer can also be used as an anti-

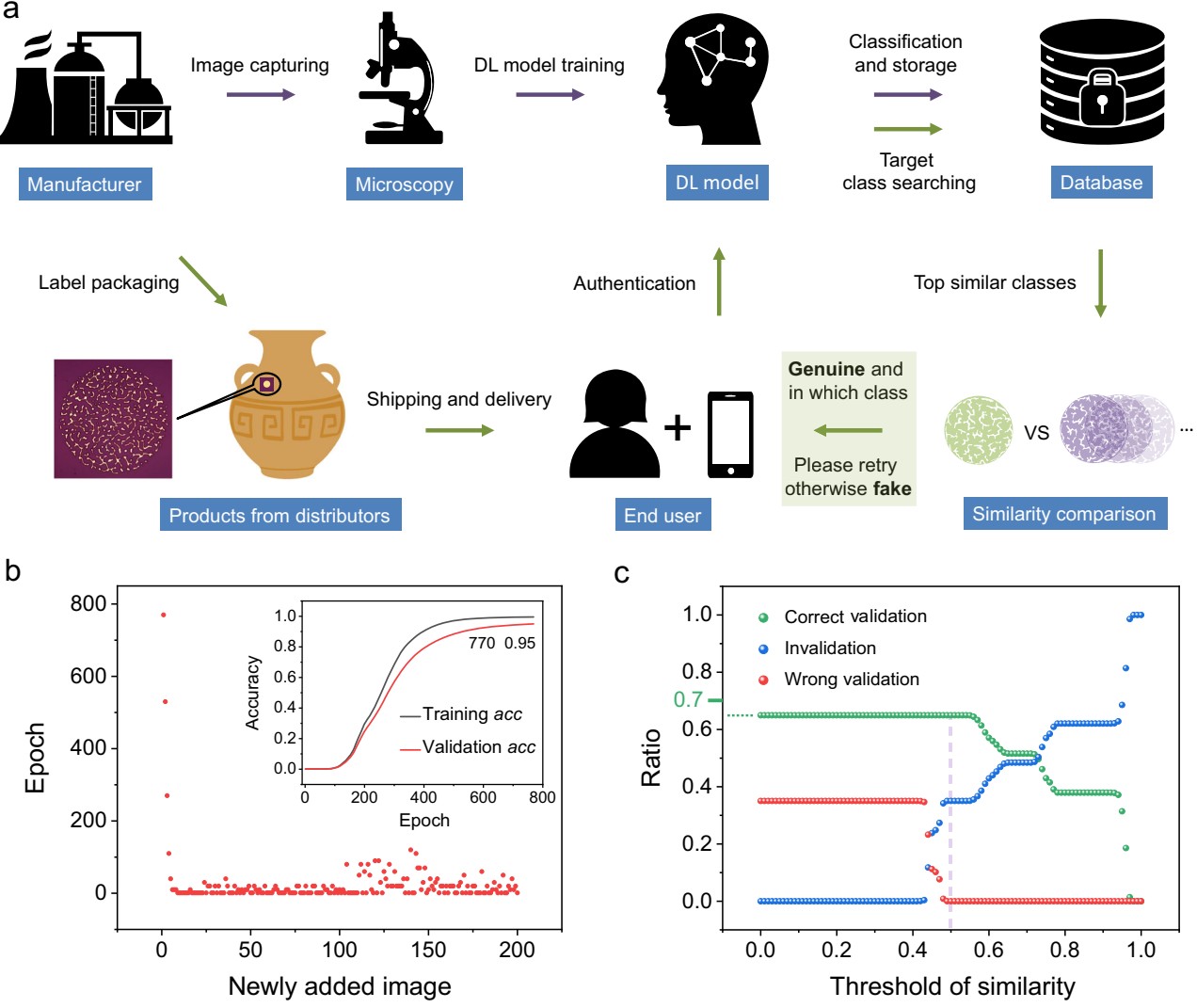

**Fig. 4 | Deep learning-based authentication system and anti-counterfeiting strategy. a** Conceptual schematic of the product authentication flow. Two pipelines are used throughout the entire process: (i) "Registration": For manufacturers, they are mainly responsible for training the deep learning model and building up the key database; (ii) "Authentication": For end users, they only need to capture the image and upload it to the database for decoding. **b** The training epoch of the classification model as a function of the newly added PUF patterns, showing a gradually stable epoch below 40. The inset shows a plot of the training and validation accuracy of the classification model with the first newly added image. The training stops when the validation accuracy reaches the threshold of 0.95 (within 770 training epochs), and *acc* represents the accuracy. **c** Ratio of correct validation, wrong validation, and invalidation of the 37,000 images as a function of similarity threshold. The 0% wrong validation can be achieved beyond the threshold of 0.5. 0.7 is the ideal maximum ratio of correct validation, as the whole test dataset contains 30% fake images.

transfer protective layer to avoid malicious replication, synergy with the potential chemical encoding (also has the long-term stability in daily light, see details in Supplementary Fig. 18). Microfabrication of similar network structures is theoretically possible, but the cost of a mass duplication of "fingerprint" labels and the intricate manufacturing technology with nanoscale accuracy eliminates the risk posed from such a laborious endeavor. In addition to the environmental stability of the security labels, mass production, and low cost are also the primary considerations in commercial applications[1]. During the high-throughput production process of the proposed labels, a mass of the PUF tags is estimated to be simultaneously integrated on a wafer by a combination of mask-assisted UV lithography (alternatively, a stencil mask) and film deposition as well as one-step thermal annealing technique (Supplementary Fig. 19) followed by wafer cutting and product packaging compatible with microchip fabrication, meeting the needs of scalability and mass production in the industry. Each PUF label is estimated to cost only approximately US$ $7 \times 10^{-3}$ (see

calculation details in Supplementary Note 2), demonstrating their commercial feasibility in economics.

## Authentication of the PUFs

Because the graphical PUFs can be easily read out using an optical microscope, a deep learning-based image identification system is appropriate to perform the tag authentication. However, once a new key is added, a separate model must be retrained, thereby resulting in a long training time with a large database. We thus proposed an improved deep learning-based authentication system with a dynamic database strategy.

Figure 4a schematically demonstrates the deep learning-based authentication system of the security label. First, each PUF pattern is captured by the manufacturers using a microscope. The different images are then preprocessed by modification of the grayscale distribution (Supplementary Fig. 20) and devoted to a ResNet50-based classification model[54] for learning the characteristics of the PUF

patterns. To reduce the training time and guarantee the generalizability of the model, the pretrained model parameters on ImageNet[55] are used to initialize ResNet50. Next, the images are classified in a general manner and stored in the database for subsequent authentication. Regarding the further database expansion, to avoid training a separate model for each image and to shorten the training period, we proposed a dynamic database strategy, and only the final layer of the classification model needs to be trained with new images added. The procedures described above are performed by the manufacturers (see details in Supplementary Note 3 and Supplementary Fig. 21). Meanwhile, the produced PUF labels are sold to goods producers for packaging and delivered to consumers through commodity circulation with rounds of authentication. The graphical security tags can be conveniently read out by the end users using their smartphone connected with a portable mini-microscope (Supplementary Fig. 22) and then uploaded to the deep learning engine for authentication. The preliminarily predicted target images with top similarities (outputs from the trained deep learning model) are searched from the database, and then, the similarity indexes between the target images and the uploaded image are compared by FSIM to further select the most similar image to achieve a higher validation accuracy. If the selected highest similarity is higher than the setting similarity threshold, the image is identified as the genuine code, and the software feeds back on the category that the image belongs to, which can be used for product traceability and anti-channeling (i.e., giving the information of product source and product distributor). Otherwise, if the similarity is lower than the threshold, the captured image cannot be validated, meaning that the image is preliminarily considered not in the established database (see more evaluation details in Supplementary Note 3). Then, the client is reminded to capture the original image again with a better imaging quality for further identification. This process can intelligently eliminate the incorrect authentication caused by poor capturing quality.

To experimentally demonstrate the above authentication system, 1300 different PUF tags were randomly captured to establish the security key database (parts of PUFs shown in Supplementary Fig. 23). Totally, 1100 PUFs were used to establish a basic database, and the remaining 200 PUFs were used for the key expansion test of the database. Each of the 1100 PUFs was rotated with different angles, forming 211,200 images as the dataset for deep learning model training (13,200 images for training and 198,000 images for validation, see details in Methods). Every input image is preprocessed by grayscale stretch to adapt to the influences of different brightness/contrasts, and random noise is added to avoid overfitting. This training process was repeated until convergence (a total of 2500 epochs), and the model at the epoch of 2250 with the highest validation accuracy (99.63%, Supplementary Fig. 24) was selected as the final base model. In the database expansion test, when the fifth new image was added, the training was finished in 5.48 s (a total of 40 epochs, 0.137 s for each epoch, Fig. 4b). Another 195 images were successively added to the deep learning model, and a gradually stable training epoch below 40 was demonstrated (Fig. 4b), allowing for a large number of new PUFs and a low time cost. This strategy helps to break the trade-off between the large key database of the PUF system and the long training time of the deep learning model in practice.

To test the proposed authentication system, 26,000 images from above 1300 PUF tags (in the database) under different conditions (brightness, rotation angle, and random noise, Supplementary Fig. 25) and 11,000 images from 550 new PUF tags (not in the database) were captured and uploaded to the trained AI for a test. We investigated the validation ratio as a function of the similarity threshold to obtain the best threshold of validation. Figure 4c reveals that when the similarity threshold reaches 0.5, we can achieve a 0% wrong validation ratio of the genuine images (i.e., the rate of false positives is 0%) and 5% false negatives, demonstrating the good effectiveness and adaptability of

the proposed authentication system. In this study, we set the similarity threshold at a value of 0.5, and the authentication of a PUF tag with an encoding capacity of approximately $10^{348}$ can be finished in 6.36 s.

## Discussions

We have developed an effective anti-counterfeiting system comprising random fractal network-based PUFs and a deep learning-based authentication strategy. The complexity of the Au network tag can be flexibly regulated by changing the film thickness, allowing the overcomes of the trade-off between the high code configurability and relatively low stability in general PUF taggants. A comprehensive understanding of the smart code control mechanisms for the thin film annealing process is also presented. Finally, based on a large basic database composed of 1300 PUF tags, a dynamic deep learning-based authentication system with an expandable database is proposed for the reliable (0% false positives), rapid, and traceable decoding of PUFs (about 37,000 images for authentication).

The conceptual presentation of the SERS-based encoding mode reveals that the Au network-based PUF can be used as a universal plasmonic platform and has the potential for carrying more information and further improving label security. Multidimensional chemical encoding can strongly fight against sophisticated forgeries but also requires a long readout time and special readout conditions. However, with the rapid development of the handheld Raman system with a high-speed readout[24], Raman characterization has the potential to be a convenient readout way in the near future.

In contrast to other physical or chemical graphic-based PUF manufacturing strategies, fractal-guided surface evolution of the thin film is an emerging and universal methodology in PUF design due to its inherent uniqueness and universality of materials. In addition to the described Au film annealing technique, diffusion-limited aggregation growth, such as electrochemical deposition, metal-induced crystallization, or evaporation-driven crystallization, can also be exploited as effective approaches in designing on-demand PUF keys for anti-counterfeiting or data encryption in the future. Although we focused on the Au network as the PUF carrier in this study, this sophisticated and universal self-organization process can be extended to various material systems at multiple scales and is anticipated to be developed with diverse matrix-dependent functional integrations, such as flexible packaging or invisible displays. The current PUF design also leaves room for improvement to cover various product fields, such as electronics. Due to the good compatibility with the microelectronic process, the network tag induced by rapid thermal annealing (RTA) is expected to be seamlessly integrated with electronics and fabricated in great batches on one wafer[44]. Overall, the proposed PUF-based anti-counterfeiting system complies with the demands necessary for commercial applications, and the proposed fractal-guided manufacturing strategy provides promising insights into the design and development of comprehensive PUFs with high code configurability, multiple-level security, environmental stability, and mass production.

## Methods

### Materials

Gold target (99.999%) was purchased from ZhongNuo Advanced Material Co. Ltd. Poly(methyl methacrylate) (average Mw -996,000 g mol$^{-1}$) and rhodamine 6G were all purchased from Sigma–Aldrich. Photoresist (S1813) was purchased from Microresist Technology GmbH, and photoresist (Ar-3110) was purchased from Allresist. The photomask was purchased from the Institute of Microelectronics of the Chinese Academy.

### Fabrication of the random Au network-based PUFs

The fabrication flow of the random Au network-based PUF tag is illustrated in Fig. 1a. In the typical procedure, a commercially available silica-coated silicon wafer (silica thickness = 300 nm) was cut into

small pieces of 1.5 cm × 1.5 cm and then cleaned with acetone, ethanol, and deionized water (DIW) successively. The cleaned substrate was spin-coated with a layer of photoresist (Ar-3110) (1 min at 3000 rpm) and baked on a hotplate at 90 °C for 1 min. Then, a circular area with a diameter of 120 μm was exposed via raster scanning by using a laser direct writing system (HWN LDW-LM4, laser wavelength = 405 nm, laser peak power = 83 mW, pulse duration = 2000 ns, and scanning frequency = 100 Hz). The exposed sample was immersed in the developing solution for 100 s for the as-prepared patterns. A layer of the Au film was deposited on the prepared substrate by magnetron sputtering (Kurt J. Lesker PVD75, DC power = 50 W, and Ar flow rate = 23 sccm). The 70-nm-thick Au film can be obtained by deposition for 1440 s. After the photoresist was lifted-off with acetone, the Au circle patterns were obtained. The sample was then annealed in a muffle furnace (QSX1200) in the atmosphere at 800 °C for 90 min, followed by cooling to room temperature to obtain the PUF tags. Network-based PUF tags can be simultaneously obtained on the support via the facile and high-throughput one-step annealing technique. Finally, the sample can be spin-coated with a thin layer of PMMA (40 s at 5500 rpm) to prevent the PUF tag from various external damages. Various PUF tags in sizes and shapes can be simply realized by laser lithography. For the configurable network tag design, Au films with different thicknesses of 30, 50, 70, and 90 nm were deposited for 540, 900, 1440, and 1800 s, respectively. The corresponding annealing parameters were 30 min at 500 °C, 90 min at 700 °C, 90 min at 800 °C, and 90 min at 900 °C. The annealing time is not a restricted parameter; a marginal change around the given time is acceptable. The fractal dimensions of the Au networks from different thicknesses were counted by using the method of box counting in the fractal theory via ImageJ software with the FracLac plugin[50].

### Implementation of PUFs with multiple-level security

The annealed Au network was first roughened by a plasma cleaner (HM-Plasma5L) for 210 s at a power of 70 W to create a plasmonic matrix. A small dose of rhodamine 6G ethanol solution (1 mM, 10 μL) was dropped on the plasmonic network, which was then washed with ethanol and DI water after full evaporation for the Raman test. The graphics of "BUAA" and QR code were fabricated by mask-assisted UV ultraviolet lithography under hard contact mode (MJB4, SUSS Micro-Tec, exposure time = 48 s) followed by the deposition of the 70-nm-thick Au film and annealing for 90 min at 800 °C. The electromagnetic-field distribution of the roughened surface of the Au network was simulated by commercial Lumerical FDTD Solutions software. Surface nanostructures were modeled in a simulation domain of 120 nm × 120 nm × 60 nm. A high-resolution simulation (mesh size of 0.5 nm) was run at a 514 nm excitation utilizing a p-polarized incident plane wave.

### Environmental stability tests

The Au network tags were characterized by optical microscopy before and after the exposure under various harsh conditions. The detailed steps are described as follows:

Low temperature: the PUF label was put into the freezer dryer (SCIENTZ-10N) at −40 °C for 60 h.

High temperature: the PUF label was put into the muffle furnace at 600, 650, 700, 750, and 775 °C for 10 h, respectively.

Aqueous corrosion and sonication: the PUF label covered with the PMMA layer was put into the DIW and sonicated for 10 min at a working frequency of 40 kHz.

Mechanical abrasion: the PUF label covered with the PMMA layer was rubbed on the frosted desktop ten times.

Dust and stain exposure: the PUF label covered with the PMMA layer was exposed to sand, and organic contamination, respectively, followed by wiping with an ultra-clean cloth dipped in alcohol.

Photostability of the chemical encoding: the Raman spectra of the identical PUF label that was placed in real-world conditions with normal ambient light exposure, oxidation, temperature, and humidity were collected again six months later under identical collecting parameters.

All the contrast images are shown in Supplementary Figs. 14–18.

### Characterization

Optical microscopic bright- and dark-field images of the random Au network morphology were captured by a polarizing microscope (Olympus, BX53M, magnification: 50×). The 3D microscopic image of the network tag was obtained by a laser scanning confocal microscope (Olympus LEXT-OLS4000, magnification: 50×). The image of the network tag in the readout demonstration was captured by a smartphone (HUAWEI nova 6) connected with a mini-microscope (commercially purchased). The X-ray diffraction data of the Au film before and after annealing were measured by an X-ray diffractometer (D8 ADVANCE). The 3D topographic image of the roughened Au network surface was obtained by AFM (Bruker Dimension Icon). Raman spectra and mapping of R6G were collected by a laser Raman spectrometer (Renishaw inVia plus) with a 514-nm laser (1.4 mW power at the sample surface) focused by a 50× microscope objective lens with a 10 s exposure time and one accumulation. The spectral data and Raman mapping were analyzed by the WiRE 2.0 software suite. The morphology contrast of the network surface before and after ion cleaning was obtained by high-resolution scanning electron microscopy (HRSEM) (Hitachi, S-8200). The thicknesses of the Au film and PMMA film were measured by a Step Profiler (DEKTAK 6 M). The amplitude and wavelength of the random networks were measured and counted by measuring software in the laser scanning confocal microscope and further verified by the Step Profiler.

### The deep learning-based authentication system

Based on the PyTorch library, a CNN-based deep learning model was built to identify and classify the PUF tags (Supplementary Fig. 21). The developed codes were run in PyCharm 2020.2 using a GPU (GeForce RTX 2080 Ti) to perform the calculations. The uploaded PUF images were first preprocessed by the localization of the PUF center (Otsu's method) and graying (Supplementary Fig. 26). Given that the images obtained by the microscopy may have different brightness or contrasts, the captured images were implemented with grayscale stretch by the homemade algorithm for unifying the image information (Supplementary Fig. 20). Therefore, the influences of these different conditions can be suppressed in deep learning model training. The captured images were then resized for training. Considering the imaging difference from different consumers, several images in different conditions were also obtained from one PUF tag (Supplementary Fig. 25) to test the trained AI model.

Randomly rotated images (rotated from 0° to 330° at 30° internal, a total of 12 images for each pattern) were involved in the deep learning model training for implementing the data augmentation to obtain the general neural networks. Rotated images from 1° to 359° at 2° internal, a total of 180 images for each pattern, were used to validate the trained model at each training step. More specifically, 1100 PUF patterns were used to form the base dataset consisting of 211,200 images in this study. 13,200 images among them were used as the training set (12 × 1100 images), and 198,000 images were used as the validation set (180 × 1100 images). All dataset information for the AI model training/validation/testing is shown in Supplementary Table 1. Random Gaussian noise was added to the training image to avoid overfitting. After each training epoch, the images used for validation were sent to the validation procedure of the model. This training process was repeated until convergence. The training images can then be removed because they do not need to be stored. Therefore, the storage space is only determined by the model parameters as well as

the initial images for each pattern. The ResNet50-based classification model was trained within 2500 epochs, where each epoch required 51.4 s of computation time. Therefore, the base deep learning training model takes approximately 35 h.

A ResNet50-based classification model that was pretrained by ImageNet[55] (3.2 million images) is used as the classifier (see details in Supplementary Note 3). To reduce the training time and guarantee the generalizability of the model, the pretrained model parameters on ImageNet are used to initialize ResNet50. With a new PUF, only one parameter is added to the last classification layer, and only this final layer is updated during the training of the new PUF. Both the original PUFs in the database and newly added PUFs are used to update the model to avoid the model forgetting the previous database, and the training of the updated model stops when a validation accuracy threshold of 0.95 is reached. The update of only the last layer ensures a fast training procedure (0.137 s per epoch with the fifth new PUF added). Owing to the "black box" mode of deep learning, it is advantageous for PUF anti-counterfeiting as it is a tamper-proof.

### Ethics and inclusion statement
This work does not include any local researchers throughout the research process and any studies with animals performed by authors.

### Reporting summary
Further information on research design is available in the Nature Portfolio Reporting Summary linked to this article.

## Data availability
The data needed to evaluate the conclusions are present in the paper and the Supplementary Information. The PUF images with corresponding datasets used for the PUF authentication have been deposited in the repository https://github.com/Seven-year-promise/PUF_authentication. Additional data related to this paper are available from the corresponding authors upon request. Source data are provided in this paper.

## Code availability
All codes used for the PUF authentication are available via https://github.com/Seven-year-promise/PUF_authentication.

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

## Acknowledgements

We gratefully acknowledge the financial support from the National Natural Science Foundation of China (NSFC) (No. 11774018 and No. 12074024, Z.Y.C.; No. 51971070 and No. 10974037, Q.L.), National Key Research and Development Program of China (No. 2016YFA0200403, Q.L.), Eu-FP7 Project (No. 247644, Q.L.) and the Natural Science Foundation of Beijing, China (No. 2202026, Y.X.).

## Author contributions

N.F.S. conceived the project. Q.L., Y.X., and Z.Y.C. supervised the project. N.F.S. performed the whole experiment, from the PUF label fabrication to performance characterization. Y.K.W. carried out the feature similarity algorithm and deep learning model training/testing. S.W. performed the mask-assisted UV ultraviolet lithography. N.F.S. wrote the whole paper. Q.L., Y.K.W., and Y.X. revised the paper. All authors reviewed the paper.

## Competing interests

The authors declare no competing interests.
