## [Peer Review File · Nature Communications]

Random fractal-enabled physical unclonable functions with dynamic AI authenticationReviewers' comments:

Reviewer #1 (Remarks to the Author):

The authors present a random fractal-enabled PUF based on non-deterministically formed gold network patterns. Authentication is performed using graphical/imaging techniques aided by a dynamic deep-learning system and Raman signatures are shown as an added layer of uniqueness/complexity.

While I enjoyed reading this manuscript and find the investigation very thorough, and of high quality scientific rigor, several issues/concerns currently preclude recommending this work for publication in Nature Communications.

Regarding the PUF:

1A.) The authors claim “We have developed an unbreakable anti-counterfeiting system” which relies on a complex Au pattern/topography. However it's not clear why such a system is unclonable. It seems that an attacker could simply take an image of the tag and intentionally fabricate their own copy of that particular pattern. Assuming the Response images are binarized as indicated in Supplementary Figure 9, it seems there is a clear vulnerability to the proposed platform. Even if the response images are not binarized, but maintained in grayscale to maintain a high encoding capacity it's not clear if the proposed approach would reject a sophisticated forgery while still authenticating genuine devices.

1B.) There does not appear to be much “grayscale” information in each tag — most of the features seem to be binary in nature (gold vs. no gold). From this perspective the present work does not seem like a significant advancement over prior literature (References 7-11) where wrinkling/crumpling provide features with dynamic grayscale information. To merit publication in Nature Communications, this work would need to present a substantial advance relative to the prior art — however, such an advance does not appear to be present or convincing.

2.) While the nanoscale roughness and Raman activity do add enhanced security — and likely the Raman map could not be cloned — the primary focus and most scalable implementation appears to be on the image based authentication. Despite the increased prevalence of handheld Raman readers — a microscaled Raman mapping is significantly slower, more costly, and more challenging to implement in practice. The reliability of the Raman enhancement would also become a concern to address.

Regarding the AI authentication:

3.) The setting of the model training is not convincing. The model architecture shown in Supplementary Figure 19 is over-parameterized for a training dataset of 2700 images. The choice of this architecture needs to be justified.

4.) 100% validation accuracy is typically a red flag in machine learning applications. It is likely that the validation data do not separate well from training data. If the training accuracy is also high, the model is highly overfitted, which is actually expected under such an over-parameterized model. As a result, the model would not have high generalizability.

5.) The proposed dynamic database strategy is similar to the method in incremental learning or online learning. It is unclear whether the new images or the entire dataset is used to fine-tune the FC layers. Nevertheless, it is well known that such a method may suffer from catastrophic forgetting, especially when you start with only 20 PUF tags. The authors are recommended to show the test accuracy of the first 20 PUF tags after adding a large number of tags, e.g., another 20, to show the effectiveness of the proposed method.

6.) Conversely, using AI-based authentication indicates that the PUF can be modeled. Discussions

are required regarding how the designer can ensure the machine learning model cannot be captured by the adversary and then used to compromise the authentication systems, especially given that autoencoder can be easily used as a generative model.

Reviewer #2 (Remarks to the Author):

The paper proposes a method to fabricate unclonable tags for anti-counterfeiting, based on the annealing of Au thin films of various thickness (being the major parameter to control the image properties of the tag). This work belongs to an ever increasing group of papers targeting at methods for fabrication of unclonable tags for object authentication. In fact one could classify these methods in those based on some kind of fluorescence after being irradiated with laser sources of proper wavelength and those exhibiting a static image with unique properties. The approach proposed by the paper belongs to the second category. The basic methodology for characterizing a "PUF" is statistical, targeting at evaluating its performance in terms of "robustness" and "unclonability" as discussed in many published papers on PUFs. Robustness is evaluated by interrogating the random structure under all different conditions that could take place in reality (illumination, temperature, surface dirt, any kind of surface degradation, etc.) and calculate the correlation of the obtained images, while unclonability is evaluated by calculation the cross correlation of different tags generated under different fabrication conditions.

Since it is a statistical analysis, it requires a very large number of samples (e.g. at least a few tens of thousands). This is my major objection with this paper. The authors fabricated only 20 samples, a number absolutely insufficient to evaluate their structures as "PUF" as it appears in fig. 3b of the paper.

Moreover, the authors provide a number of claims to support the random nature of the fabricated tags. Starting with the major claim of "fractal structures" of the formed patterns (called "networks") in the paper, from the figures of the fabricated patterns they comprise of dendrite structures with different number of branches but to my opinion these patterns do not exhibit statistical self similarities between local and whole geometries appearing at different scales as defined by Mandelbrot in his pioneering work (ref. 27 of the paper). And in any case no mention should be made on "Chaos" and "Chaotic traits" (page 64,65 of the paper).

Going to the calculation of the encoding capacity of 10630 (line 243 of the paper) this comes from the assumption that two different "PUFs" can differentiate by one pixel (out of 10⁶) and one grayscale tone (out of 255). Obviously, this number is totally unrealistic taking into account the actual properties of the imaging unit (optical interrogator) in terms of modulation transfer function of the optics or the noise of the detection unit, etc. Moreover such minor differences would result in very high cross correlation between the corresponding patterns, reducing radically the number of usable uncorrelated tags.

Concerning the authentication method, as the authors mention, a conventional image processing analysis could be used to extract the image features of the tag. The authors have developed a modified deep learning scheme for this purpose. Here also the paper suffers from the negligible number of samples. For example, for the generation of the training set they used fifteen out of the twenty sample patterns and they enriched them with additional patterns coming from rotation of the initial 15 patterns by one degree. I'm not sure if this is the best training approach but again a deep learning system based on a total of 20 samples is out of the question. Moreover, I'm wondering what are the benefits of the ML method compared to the conventional image processing methods.

Concluding, I cannot recommend publication of the submitted paper to Nature Communications.

Response Letter

We are grateful for the positive comments and helpful suggestions from the reviewers. Our responses to the reviewers' comments and the changes made in our manuscript are as follows. For the sake of readability, our responses are highlighted by blue font and the changes to our manuscript are presented in green.

Reviewer #1:

The authors present a random fractal-enabled PUF based on non-deterministically formed gold network patterns. Authentication is performed using graphical/imaging techniques aided by a dynamic deep-learning system and Raman signatures are shown as an added layer of uniqueness/complexity.

While I enjoyed reading this manuscript and find the investigation very thorough, and of high quality scientific rigor, several issues/concerns currently preclude recommending this work for publication in Nature Communications.

We truly appreciate the reviewer for the careful consideration the enthusiastic support of our manuscript.

C1: Regarding the PUF:

1A.) The authors claim “We have developed an unbreakable anti-counterfeiting system” which relies on a complex Au pattern/topography. However it's not clear why such a system is unclonable. It seems that an attacker could simply take an image of the tag and intentionally fabricate their own copy of that particular pattern. Assuming the Response images are binarized as indicated in Supplementary Figure 9, it seems there is a clear vulnerability to the proposed platform. Even if the response images are not binarized, but maintained in grayscale to maintain a high encoding capacity it's not clear if the proposed approach would reject a sophisticated forgery while still authenticating genuine devices.

R1: We thank the reviewer for proposing this general issue. We really agree with the reviewer that the attacker can pose a great threat to PUF security. First and foremost, due to the trait of mass production in the proposed PUF fabrication, each “fingerprint” tag is allowed to only protect and identify one particular good. Microfabrication (e.g. micro/nano 3D printing technique) of the Au network structures is theoretically possible, but the time and equipment needed to duplicate a mass of anticounterfeiting labels eliminate the risk posed from such a laborious endeavor. Compared with the economic value of the protected mass-market products, the cost spent on the duplication technology makes the duplication itself a bad bargain.

Additionally, the three-dimensional height information of the network encoded into the PUFs can also be used for anti-clone, while the height information can't be simply obtained by general imaging. The random surface texture and height fluctuation at the nanoscale also increase the difficulty of the full duplication of grayscale details. Although remarkable progress has been made

in 3-dimensional micro/nano fabrication technologies from now, elaborate architecture at the nanoscale and flexible employment of host materials still remain a big challenge. Noting that the binary images in Supplementary Figure 9 are only used to extract the physical feature and count the filling ratios for encoding capability, rather than to be regarded as the security key for authenticating the goods. The security key can be maintained in grayscale and carried more information, more details are shown in **Comment 2**.

Moreover, the conceptually demonstrated chemical tag orthogonal to the physical feature in our manuscript enables the network to be a multidimensional PUF tag with a higher non-replicability. The fabrication of the plasmonic platform can also increase the difficulty of the duplication for attackers. Potentially, an upgraded encryption strategy can be carried out by introducing a random loss of signal hid in the chemical pattern, as shown in Figure R1. By artificially blocking random positions of the physical pattern, the orthogonal chemical pattern would be incomplete. Therefore, even if the physical object has been accurately duplicated by a sophisticated forgery, the corresponding chemical identifier still remains an unpredictable variable (intrinsic information preserved by manufacturers), which can be also responsible for anticounterfeiting. In other words, an accurately replicated chemical pattern can be regarded as the fake instead, guaranteeing an almost unbreakable anticounterfeiting system. Finally, it's worth noting that the developed PUF is not only a 3D physical object in visual, but also a hierarchical complex structure extending from microscale to nanoscale and integrated with multiple responses. This makes the duplication of the hierarchical PUFs an almost impossible project.

Figure R1. Upgraded chemical encryption.

Regarding the authentication system, we can implement the one-time authentication strategy at the user side. After one authentication, the security key can be permanently invalid in the secure server and the security key can be removed from the database, preventing a cloned PUF from being reused.

Actually, sophisticated forgery remains a common issue for all kinds of PUF labels. Our article mainly demonstrates the concept and methodology of the fractal-enabled PUF system, so we do not make many assumptions about possible loopholes in real-world applications.

“Microfabrication of similar network structures is theoretically possible, but the cost of a mass duplication of “fingerprint” labels and the intricate manufacturing technology with nanoscale accuracy eliminate the risk posed from such a laborious endeavor.”

“The conceptual presentation of the SERS-based encoding mode reveals that the Au network-based PUF can be employed as the universal plasmonic platform and has the potential for carrying more information and further improving the security level of the label. Multidimensional

chemical encoding can strongly fight against the sophisticated forgery...”

C2: 1B.) There does not appear to be much “grayscale” information in each tag — most of the features seem to be binary in nature (gold vs. no gold). From this perspective the present work does not seem like a significant advancement over prior literature (References 7-11) where wrinkling/crumpling provide features with dynamic grayscale information. To merit publication in Nature Communications, this work would need to present a substantial advance relative to the prior art — however, such an advance does not appear to be present or convincing.

R2: Thank the reviewer for the important and detailed comments. The response image of Si/SiO₂ support is truly in a single grayscale, which has been excluded from the encoding capacity calculation. However, our PUFs are not the regular structures, different from the structures produced by the template-assisted film deposition. Oppositely, the PUFs are based on the three-dimensional Au networks spontaneously shrinking from the Au film, in which the curved edges, height fluctuation of the rugged surface (Fig. 2e, f and g in the manuscript), random textures and other topography difference can generate rich information in contrast (grayscale) on the response images due to the different reflection and refraction of light. Therefore, the ridge-like morphology of the Au network is actually similar to the polymer wrinkle, as compared in Figure R2. Moreover, from the grayscale distribution (Figure R3) of one Au network-based PUF, we can see that the response image dose presents a uniform full-grayscale distribution from 0 to 255. As mentioned, it seems to be only two grayscale values (gold vs. no gold) in the response image with the naked eyes. Therefore, we carried out the presentation of grayscale distribution through algorithm. Although the subtle grayscale distribution is difficult to be distinguished with the naked eyes, this can be actually recognized by the algorithm accurately and also be responsible for the credible encoding capacity calculation.

Figure R2. a-b The ridge-like morphology of the Au network. c The ridge-like polymer wrinkle (*Adv. Funct. Mater.* 31, 2106754, 2021).

Figure R3. The full-grayscale distribution from 0-255.

Besides, we also realize that the 1000×1000 pixels of the PUF feature image may be relatively high so that the grayscale information will be unobscure. We can flexibly change the different size of the captured images to highlight the grayscale information, although the corresponding encoding capacity will be decreased. The contrast of response images with different sizes (sizes from 1000×1000 to 250×250) is shown in Figure R4. Here we set the size of a single PUF feature image to 750×750 for showing a more evident and rational response image in grayscale. Meanwhile, the encoding capacity proportional to the image size can be decreased correspondingly (from 10^{630} to 10^{353}), while is still beyond the standard encoding value of 10^{20} . Notably, a lower size of image is not desirable, because it will obscure some feature details.

Figure R4. The response images with different sizes (left to right, 1000, 750, 500 and 250). One physical feature is extracted as a demonstration.

We really agree with the reviewer that the wrinkling/crumpling due to the compressive stresses in layered systems can provide more abundant topography feature information than our network-based PUF tags, which can provide a higher encoding capacity. However, a PUF with extremely large encoding capacity can reduce the probability of fabricating two tags with same configuration by a stochastic process and resulting in the same response, which can cause a low reproducibility and readout robustness. Besides, the PUF with dense grayscale information is also hard to be applied to cryptographic primitives owing to their ultrahigh complexity and poor stability.

The encoding capacity is not the only concern of a PUF system and the focus should not be blindly pursuing a high encoding capacity. The robustness of the PUF system is also the major consideration. In fact, our PUF tag is specifically designed aiming at the shortcomings existing in conventional graphical PUFs, especially the wrinkling/crumpling PUF system. The advances can be shown as follows,

1) our Au network-based PUF has inherited the high configurability in tag complexity of the flexible wrinkling system, such as the on-demand design on dimensions of physical features and external geometric of the single tag, while having a higher physical robustness in extreme conditions (such as the high temperature durability reaching 800°C) benefits from the inert-metal's intrinsic stability. The elastomeric polymer often suffers from the physical aging in high temperature, humidity/water, or oxygen. This breaks the trade-off between the high code configurability and relatively low stability in general PUF taggants, which is the main advances compared with the prior art.

2) the high-throughput production process combined with the mask-assisted UV lithography and film deposition ensures the mass production of the labels. Besides, a better compatibility with the microelectronic process along with a high code configurability and miniaturization broadens the application fields of the PUFs in hardware products.

3) our PUF can be regarded as the inherent and homogeneous plasmonic platform for multiple response and multiple-level security, which is independent of chemical-synthesis nanoparticles and complex surface modifications necessary for other wrinkling/crumpling system (*ACS Appl. Mater. Interfaces* 8, 4031–4041, 2016. *ACS Appl. Mater. Interfaces* 13, 11247–11259, 2021).

4) the fractal-guided manufacturing strategy also provides a new perspective into the concept design of PUFs and can be universally extended to various material systems at multiple scales.

“The 3-dimensional feature information can be encoded into every pixel of the PUF pattern, while each pixel is utilized as variables in different grayscale levels (grayscale levels in the range of 0-255, which is presented in a uniform full-grayscale distribution in Supplementary Fig. 8) derived from the structure height-dependent different light reflection and refraction.”

“The encoding capacity of the PUFs is not dominant compared to wrinkling/crumpling system with dynamic grayscale information or random distributed stimuli-responsive taggants with multiple responses. However, a PUF with extremely large encoding capacity can reduce the probability of fabricating two tags with same configuration by a stochastic process (i.e., a low reproducibility) and resulting in the same response. The ultrahigh complexity and poor stability also cause the difficulty of the application in cryptography.”

C3: 2.) While the nanoscale roughness and Raman activity do add enhanced security — and likely the Raman map could not be cloned — the primary focus and most scalable implementation appears to be on the image based authentication. Despite the increased prevalence of handheld Raman readers — a microscaled Raman mapping is significantly slower, more costly, and more challenging to implement in practice. The reliability of the Raman enhancement would also become a concern to address.

R3: Thanks for pointing out this concern. As the reviewer said, Raman detection especially the micro-scaled Raman mapping dose has shortcomings in convenience and efficiency. Importantly, the readout time and corresponding readout hardware, as well as the security level of the security key should be commensurate with the economic value and purpose of the product that the tag protects. Actually, this additional chemical encoding mode can be potentially applied to the tailored products, such as the jewelry or antiques. The owner of a luxury product can have a customized authentication during purchase by a staff with handheld Raman readers or even sending to a laboratory.

Recently, many studies have reported about the ever increasing spectral PUFs based on the

Raman mapping (*Nano Today* 41, 101324, 2021. *ACS Appl. Mater. Interfaces* 13, 11247–11259, 2021. *Nanoscale* 12, 9471–9480, 2020. *Nat. Commun.* 11, 516, 2020. *ACS Appl. Mater. Interfaces* 8, 4031–4041, 2016). The concern of the reading speed has been paid more and more attention by researchers. For example, Yuqing Gu et al. reported a gap-enhanced Raman PUF tags using a high-speed Raman mapping method, which can achieve a Raman mapping-based PUF tag ($100 \times 100 \mu\text{m}^2$) readout in DuoScan mode in only 6 s with a resolution of 50×50 pixels (*Nat. Commun.* 11, 516, 2020). Nevertheless, the scanning speed of the lab-based confocal Raman system will be further continuously improved for practical use with many strategies and even the Raman mapping can be potentially realized on a hand-held device in the future. Apart from the synthesis of Raman tags with better performance, new Raman imaging modes could be introduced, such as the application of line-shaped (*Proc. Natl Acad. Sci. USA.* 110, 12408–12413, 2013), multipoint laser (*Anal. Chem.* 88, 1281–1285, 2015) or direct Raman imaging with a narrow-band filter (*Nat. Protoc.* 8, 677, 2013), or a mode, where the stage movement, light collection and data readout occur continuously and synchronously (*Proc. Natl Acad. Sci. USA.* 110, 12408–12413, 2013). In brief, the engineers and industries can help apply it to the products. Actually, our article mainly demonstrates the concept and feasibility of multiple-level security strategies, so we do not have many discussions on the improvements and key problem tackling of general Raman imaging technique.

We really agree with the reviewers that the reliability of the Raman signal is also the main concern in Raman detection. Except for the influence of the Raman detection device, the uniform and repeatability of the SERS substrate are the prerequisites deciding the reliability of the Raman signal (*J. Phys. Chem. C* 120, 16946–16953, 2016). Regarding the nanoparticle-based SERS substrate, the spatial inhomogeneity of the nanoparticle morphology and of the nanoparticle (hot-spots) distribution are the main reasons of fluctuations in SERS intensity, which can cause the loss of the information and influence the robustness of the readout. In our work we introduce a uniform and large-area ion bombardment technique to fabricate the homogeneous SERS substrate. As shown in Fig. 3e in the manuscript, an amount of dense and uniform convex Au nanostructures are generated on the flat surface (Supplementary Fig. 11), which can be used as the “hot spots” of the electromagnetic field enhancement (Supplementary Fig. 12). The Raman mapping in Fig. 3g provides a direct insight into the uniform spatial distribution of SERS response, which shows the reliability of the Raman signal on our SERS substrate. The SERS substrate also has a high fabrication repeatability under the same ion bombardment power and time, ensuring the identical Raman signal intensity and distribution.

In particular, we would like to highlight that the additional Raman-based chemical encoding in the manuscript can be regarded as a proof-of-concept, which reveals that the Au network-based PUFs can be employed as the universal plasmonic platform and have the potential for carrying more information and further improving the security level of the label. More progress remains to be explored.

“The conceptual presentation of the SERS-based encoding mode reveals that the Au network-based PUF can be employed as the universal plasmonic platform and has the potential for carrying more information and further improving the security level of the label. Multidimensional chemical encoding can strongly fight against the sophisticated forgery but requires a long readout time and special readout condition. However, with the rapid development of the handheld Raman system with a high-speed readout²³, Raman characterization is potential to be a convenient readout way in the near future.”

C4: Regarding the AI authentication:

3.) The setting of the model training is not convincing. The model architecture shown in Supplementary Figure 19 is over-parameterized for a training dataset of 2700 images. The choice of this architecture needs to be justified.

R4: We thank the reviewer for pointing out this concern. In order to provide a convincing model, we changed two important parts of our AI authentication system:

1) The architecture of the model is changed to a ResNet50 based classification model pretrained by ImageNet (3.2 million images). The input image is preprocessed in grayscale by modification of the pixel distribution and sent to a ResNet50 based classification model. Some random Gaussian noise was added to the training image in order to limit overfitting and also for covering the influence from the image noise. To reduce the training time and guarantee the generalizability of the model, the pretrained model parameters on ImageNet is used to initialize the ResNet50 and a base model is trained via 1100 PUFs. With a new PUF, only one parameter is added to the last classification layer of the base model, and only this final layer is updated during the training of the new PUF. Noting that both the original PUFs in the database and newly added PUFs are used to update the model to avoid the model forgetting the previous database. The update of only the last layer ensures a fast training procedure. When the fifth new image was added, the training can be finished in 5.48 s (total 40 epochs, 0.137s for each epoch, Figure R5b) in the database expansion test. The results of the database expansion test are shown below.

2) A lot more PUF images are captured (1850 PUFs): 1) **1300** PUFs for the establishment of the database with 1100 PUFs for building the base database and 200 PUFs for the expansion of the database; 2) **550** PUFs as the fake PUFs out of the database and augmented for training a general model. The TEST_Set contains **37000** images augmented from the original 1850 PUFs, which considers different conditions (brightness, rotations, and noise) that can occur in practice, covering the influence from different capturing conditions and imaging unit. We truly know that not all conditions can be considered in this test, but our work aims to provide a benchmark to prove that our model and technique have the potential application in the PUF authentication.

The corresponding training/validation/testing results are shown below. According to the results, we can achieve a 0% wrong validation ratio of the genuine images (0% false positives and 5% false positives). Noting that the high image invalidation rate comes from the large ratio of the fake image (30%), which is tested as the invalidation in our model setup. Of course, the further optimization of the model for obtaining a higher accuracy is being implemented. Besides, we have made a list of the different database sizes for AI model training in other PUF works.

Two added references related to the modified model are as follows,

[1] ImageNet:

J. Deng, W. Dong, R. Socher, L. -J. Li, Kai Li and Li Fei-Fei, "ImageNet: A large-scale hierarchical image database," 2009 IEEE Conference on Computer Vision and Pattern Recognition, 2009, pp. 248-255, doi: 10.1109/CVPR.2009.5206848.

[2] ResNet50:

K. He, X. Zhang, S. Ren and J. Sun, "Deep Residual Learning for Image Recognition," 2016 IEEE Conference on Computer Vision and Pattern Recognition (CVPR), 2016, pp. 770-778, doi: 10.1109/CVPR.2016.90.

Table 1. The parameters for the AI model training/validation/testing.

Dataset	Size	Augmentation	Labels
Original images (ORI_Set)	1850 PUFs		
Training set (TRAIN_Set)	1100 PUFs from ORI_Set	Rotation: 0, 30, 60, 90, .. , 330°	Corresponding classes: 0-1099
Validation set (VAL_Set)	The same PUFs with TRAIN_Set	Rotation: 1, 3, 5, 7, .. , 359°	Corresponding classes: 0-1099
Candidate set (CAND_Set)	200 PUFs from ORI_Set	Rotation: 0, 30, 60, 90, .. , 330° for training and 1, 3, 5, 7, .. , 359° for validation	Corresponding classes: 1100-1299
Fake set (FAKE_Set)	550 PUFs from ORI_Set	1) Rotation: 10 angles are randomly selected from those that are not considered in the TRAIN_Set and VAL_Set. 2) Gaussian noise for each image 3) 10 different brightness conditions applied to each image	All labelled as "FAKE"
Test set (TEST_Set)	1850 PUFs of ORI_Set with augmentations that generate 37000 images		Corresponding classes: 0-1299 for the first 1300 PUFs in the database, and "FAKE" for the left 550 PUFs

Figure R5. **a** The training procedures of the base classification model (TRAIN_Set and VAL_Set). **b** The training time of the classification model as a function of the newly added 200 PUFs, showing a gradually stable training epoch below 40. The inset is the plot of training and validation accuracy of the classification model with the first newly added image. The training stops when the validation accuracy reaches at the threshold of 95%. R_c represents the accuracy. **c** Correct/Wrong validation ratio and invalidation ratio of the 37000 images as a function of similarity threshold. The ideal maximum of the correct accuracy is 0.7, as the whole test dataset contains 30% fake images. The 0% wrong validation can be achieved beyond the threshold of 0.5.

Table 2 The database size for AI model training in different works of PUFs.

PUF type	Database size for AI model training	Reference
Crumpling of 2D Materials	500	Matter 3, 2160–2180, 2020.
Drop-casting nanoparticle tags	1020	ACS Nano 15, 2901–2910, 2021.
Laser-induced polymer wrinkle	112	Adv. Mater. 32, 2003032, 2020.
Multi-functional nanoinks	100	Nano Today 41, 101324, 2021.
Random wrinkles	6	ACS Appl. Mater. Interfaces 13, 27548–27556, 2021.
Inkjet-printed unclonable quantum dot	6	Nat. Commun. 10, 2409, 2019.
Inkjet-printed unclonable quantum dot	10	ACS Appl. Mater. Interfaces 13, 15701–15708, 2021.
Unclonable photonic crystal hydrogels	10	ACS Appl. Mater. Interfaces 14, 2369–2380, 2022.
Our work	1300	

The further optimization of the model (for getting a higher accuracy) and testing of more images with different conditions are underway (images with different scaling and blurring are to be tested). As for the database expansion test, more new PUFs (another 300 images) are to be supplemented. This part of the works is expected to be finished in two weeks.

“Figure 4a schematically demonstrates the deep learning-based authentication system of the security label. Firstly, each PUF pattern is captured by the manufacturers by using the microscope. The different images are then preprocessed by modification of the pixel distribution and devoted to a ResNet50 based classification model⁴⁷ for learning the characteristics of the PUF patterns. To reduce the training time and guarantee the generalizability of the model, the pretrained model parameters on ImageNet⁴⁸ is used to initialize the ResNet50. Next, the images are classified in a general manner and stored in the database for subsequent authentication.”

“For experimentally demonstrating the above authentication system, 1300 different PUF tags were randomly captured to establish the security key database (parts of PUFs shown in Supplementary Fig. 22). 1100 PUFs of them were used for the establishment of the basic database, and the rest 200 PUFs were used for the key expansion test of the database. Each of the 1100 PUFs were rotated with different angles, forming 211200 images as the dataset for deep learning model training (13200 images with less rotations for training and 198000 images for validation). Every input image is preprocessed by grayscale stretch and added random noise to avoid overfitting.”

“For testing our authentication system, the above 1300 PUF tags (in the database) in different conditions (brightness, rotation angle, and random noise, total of 26000 images, Supplementary Fig. 24) and 11000 images from 550 new PUF tags (not in the database) were captured and uploaded to the trained AI for a test. We investigated the validation ratio as a function of the similarity threshold by testing multiple similarity indexes for obtaining the best threshold of validation. Figure 4c reveals that when the similarity threshold reaches 0.5, we can achieve a 0% wrong validation ratio of the genuine images (that is, the rate of false positives is 0%) and 5% false negatives. In this work, we set the similarity threshold at a value of 0.5, and the authentication of a PUF tag with an encoding capacity of about 10^{353} can be finished in 6.36 seconds.”

C5: 4.) 100% validation accuracy is typically a red flag in machine learning applications. It is likely that the validation data do not separate well from training data. If the training accuracy is also high, the model is highly overfitted, which is actually expected under such an over-parameterized model. As a result, the model would not have high generalizability.

R5: We thank the reviewer for pointing out this problem. The 100% accuracy came from that the choice of training and validation sets are too similar to each other. Therefore, we consider to make the dataset larger as shown in the above table 1. Additionally, we use less rotation angles in the training set compared with the validation set and apply the gaussian noise during training to avoid the overfitting of the model. We also use the pretrained ResNet on the ImageNet (3.2 million images) to do the initialization of the model parameters for a transfer learning. Based on those modifications and the result in Figure R5a, we believe that our model has a higher generalizability. As shown in Figure R5a, this training process was repeated until convergency (totally 2500 epochs, 51.4 s for each epoch) and the model at the epoch (2250) with the highest validation accuracy (99.63%) is selected as the final base model.

“Every input image is preprocessed by grayscale stretch and added random noise to avoid overfitting.”

“This training process was repeated until convergency (totally 2500 epochs) and the model at the epoch of 2250 with the highest validation accuracy (99.63%, Supplementary Fig. 23) is selected as the final base model.”

C6: 5.) The proposed dynamic database strategy is similar to the method in incremental learning or online learning. It is unclear whether the new images or the entire dataset is used to fine-tune the FC layers. Nevertheless, it is well known that such a method may suffer from catastrophic forgetting, especially when you start with only 20 PUF tags. The authors are recommended to show the test accuracy of the first 20 PUF tags after adding a large number of tags, e.g., another 20, to show the effectiveness of the proposed method.

R6: We totally agree with the reviewer about the forgetting problem of incremental learning. In such case, our previous dataset is not sufficient to verify whether that the model is still effective for largely increasing the size of the database. Therefore, we built a larger dataset (totally 1850 PUFs: 1100 PUFs for the establishment of the base model, 200 PUFs for the expansion of the database and 550 PUFs as the fake image out of the database) with a huge TEST_Set containing 37000 images from the 1850 different original PUFs. With a new PUF, only one parameter is added to the last classification layer of the base model, and only this final layer is updated during the training of the

newly added PUF. Noting that both the original PUFs in the database and newly added PUFs are used to update the model to avoid the model forgetting the previous database. Here we added 200 new PUFs as the database expansion test, and the expansion results and test accuracy on the TEST_Set are shown in Figure R5b and c.

“When the fifth new image was added, the training can be finished in 5.48 s (totally 40 epochs, 0.137s for each epoch, Fig. 4b) in the database expansion test. Another 195 images were further added to the deep learning model successively and a gradually stable training epoch below 40 is demonstrated (Fig. 4b),..”

“With a new PUF, only one parameter is added to the last classification layer, and only this final layer is updated during the training of the new PUF. Noting that both the original PUFs in the database and newly added PUFs are used to update the model to avoid the model forgetting the previous database, and the training of the update model stops until a validation accuracy threshold T_{add} is reached.”

C7: 6.) Conversely, using AI-based authentication indicates that the PUF can be modeled. Discussions are required regarding how the designer can ensure the machine learning model cannot be captured by the adversary and then used to compromise the authentication systems, especially given that autoencoder can be easily used as a generative model.

R7: Thank the reviewer for pointing out the issue. Apparently, the autoencoder is super similar to some generative models like variational autoencoder (VAE), and we have also the same concern. Therefore, we changed our model architecture to a ResNet50 based classification model without autoencoder. We truly know that the generalizability of the model can be decreased without the basic autoencoder training, so we built a much larger dataset and add the augmentation strategies to ensure the model is not over-parameterized. As for the concern caused by the adversary, our model has the capacity of filtering the PUFs that are out of the database. Although the adversary could capture the model architecture, it would be difficult to train the same model with the same parameters. Besides, all the PUFs captured by the users are only processed by our own trained model instead of being processed by the model from the adversary. We understand that the adversary has many ways to compromise the authentication system, but the method proposed in our manuscript is expected to be a benchmark to demonstrate the pipeline, so a more accurate and robust model would be trained to handle the problems that occur in practice.

Finally, we truly thank the reviewer for the helpful suggestions that help us change the architecture of the model.

Reviewer #2:

The paper proposes a method to fabricate unclonable tags for anti-counterfeiting, based on the annealing of Au thin films of various thickness (being the major parameter to control the image properties of the tag). This work belongs to an ever increasing group of papers targeting at methods for fabrication of unclonable tags for object authentication. In fact one could classify these methods in those based on some kind of fluorescence after being irradiated with laser sources of proper

wavelength and those exhibiting a static image with unique properties. The approach proposed by the paper belongs to the second category. The basic methodology for characterizing a “PUF” is statistical, targeting at evaluating its performance in terms of “robustness” and “unclonability” as discussed in many published papers on PUFs. Robustness is evaluated by interrogating the random structure under all different conditions that could take place in reality (illumination, temperature, surface dirt, any kind of surface degradation, etc.) and calculate the correlation of the obtained images, while unclonability is evaluated by calculation the cross correlation of different tags generated under different fabrication conditions.

We really appreciate the reviewer for the careful consideration of our manuscript and the significant guidance of the basic concept and research progress of the physical unclonable functions (PUFs). In the current tendency of excessively pursuing particular characteristics (e.g., encoding capacity) or employing diverse taggants, we mainly focus on the comprehensive performance of the PUF system, not just on the propose of a new methodology. We really agree with the reviewer that the statistical property is a key factor in evaluating PUF effectiveness and thank the reviewer for proposing this concern. Therefore, we mainly focused on this concern and revised our manuscript through sample collection, cross correlation calculation and AI model training/validating based on a large sample volume.

C1: Since it is a statistical analysis, it requires a very large number of samples (e.g. at least a few tens of thousands). This is my major objection with this paper. The authors fabricated only 20 samples, a number absolutely insufficient to evaluate their structures as “PUF” as it appears in fig. 3b of the paper.

R1: Thank the reviewer for pointing out this basic issue. We are very sorry that we haven’t employed enough samples for the calculation of cross correlation values by FSIM, which causing a poor evaluation of the unclonability of PUFs. The random fractal-enabled PUF fabrication process actually has the traits of high randomness and predictability, and the uniqueness has been examined by comparing their topography with each other through optical imaging. The similarity statistics of different PUFs is also very important in quantitative evaluation of the unclonability. Therefore, we have supplemented the fabrication batch and fabricated a larger amount of PUFs. We selected **600** (30 times larger than before) different PUFs for the calculation of cross correlation values to verify the uniqueness of the PUF tag. The corresponding heat map, distribution histogram of the similarity and corresponding PUF patterns are shown below. The statistical results further prove that our PUFs have an absolute uniqueness.

We have made a comparison of the sample number for similarity calculation in other related researches. It can be revealed that the several hundred samples are sufficient for evaluating the unclonability of the PUFs. Nevertheless, our sample number of 600 is undoubtedly the highest among the current work.

Figure R6. **a** Heat map showing the cross-correlation values. **b** Histogram showing the distribution of cross-correlation values obtained from the heat map.

Table 3 The sample numbers for cross-correlation value calculation in different works of PUFs.

PUF type	Sample number for similarity comparison	Reference
Crumpling of 2D Materials	296	Matter 3, 2160–2180, 2020.
Drop-casting nanoparticle tags	256	ACS Nano 15, 2901–2910, 2021.
Laser-induced polymer wrinkle	200	Adv. Mater. 32, 2003032, 2020.
Multi-functional nanoinks	20	Nano Today 41, 101324, 2021.
Polymer wrinkle	200	Adv. Mater. 27, 2083–2089, 2015.
Chaotic organic crystal phosphorescent patterns	480	Adv. Mater. 33, 2102542, 2021.
Random fluorescent proteins	30	Nat. Commun. 11, 328, 2020.
Randomly distributed fibers	30	Nat. Commun. 13, 247, 2022.
Drop-casting Raman tag	10	Nat. Commun. 11, 5543, 2020.
Folding of plasmonic gel	100	ACS Appl. Mater. Interfaces 8, 4031–4041, 2016.
Our work	600	

Figure R7. The 600 images for cross-correlation calculations.

C2: Moreover, the authors provide a number of claims to support the random nature of the fabricated tags. Starting with the major claim of “fractal structures” of the formed patterns (called “networks”) in the paper, from the figures of the fabricated patterns they comprise of dendrite structures with different number of branches but to my opinion these patterns do not exhibit statistical self similarities between local and whole geometries appearing at different scales as defined by Mandelbrot in his pioneering work (ref. 27 of the paper). And in any case no mention should be made on “Chaos” and “Chaotic traits” (page 64,65 of the paper).

R2: Thank the reviewer for the insightful consideration. Actually, fractal structures have a wide range from earth science (coastline, river and mountain) to experiment synthesis (electrochemical deposition, metal-induced crystallization and dielectric breakdown). One most widely studied class of fractal surface structures is that of deposited coatings. Agglomeration by diffusion, ballistic impact, and chemical or electrochemical processes often result in self-similar structures.

Percolation network/cluster is also a typical fractal structure with statistical self-similarities, such as the self-assembled fractal gold nanostructure via wet chemistry synthesis method (*Nano Lett.* 18, 3593–3599, 2018) (Figure R8a), fractal clusters in thin gold films through film deposition (*Phys. Rev. Lett.* 49, 1441–1444, 1982. *Phys. Rev. Lett.* 49, 1444–1447, 1982) (Figure R8b) and Au film annealed-induced random islands after de-percolation (*Nano Lett.* 20, 3291–3298, 2020) (Figure R8c). Percolation is a process during which unconnected clusters grow and fill a system. Percolation theory describes connectivity of objects within a network structure and the effects of this connectivity on the macroscale properties of the system, which is widely used to describe the stochastic geometry system like fractals.

Random fractal Au islands have been commonly prepared by annealing of Au films with different thicknesses below the percolation threshold (a given filling fraction of gold reaching the whole connection of the gold islands, *J. Phys. Chem. C* 117, 11337–11346, 2013. *Nano Lett.* 20, 3291–3298, 2020). The annealing of the Au film is the process of de-percolation and the isolated ramified Au networks described in our manuscript (Figure R8d) belongs to the isolated percolation clusters/networks with different percolation correlation lengths (percolation cluster dimension, L). In percolation, any clusters of the order of L or larger is argued to be a self-similar object up to length L (*Phys. Rev. Lett.* 49, 1444–1447, 1982), which means that the different percolation networks with different fractal orders and percolation correlation lengths (Fig. 1d in the manuscript) in a single PUF are self-similarities.

The construction of a Cayley tree is by straightforward iterative function. Upon each iteration, two new branches are added to each terminal branch. Similar to the Cayley tree model, the network can be regarded as an iterated quasi-Y shape object, and the branching fractal-Au networks are self-similar to itself in bifurcations (from main bifurcation to sub-bifurcation). The difference is that the percolation networks are not in precise self-similarities, which means that the branch orientations and lengths are random varied and the symmetry is broken (Fig. 1d in the manuscript). However, the multiple iterated bifurcations are in statistical self-similarities. For example, in Figure R8d, the part 1 (whole) and the derived part 2 (part) have the similar quasi-Y shape feature and iterative trend. Besides, the parts of the network have the same complexity with the whole network, both showing high randomness.

Thank the reviewer again for pointing out that the use of “chaos” or “chaotic traits” is wrong

here. The chaos in physics may refer to the dynamic process, while the fractal in mathematics pay more attention to the description of morphology and geometry. Therefore, we deleted this corresponding expression in our manuscript.

We are very sorry that we did not illustrate clearly about the concept of the percolation network, as a typical fractal structure, in the initial manuscript. We have further elaborated this theory in the revised manuscript.

Figure R8. a-c Three different kinds of percolation networks/clusters with fractal characterizations and d the approximate branching fractal model extracted from one percolation network-based PUF tag in our work.

“Fractal theory is also used to elucidate the complex surface morphology evolution of diverse thin film systems, such as the fractal-guided percolation networks/clusters of films^{28,29,30}. Percolation network/cluster refers to a system in global connectivity through a continuous “chain” of locally connected objects, such as the self-assembled Au nanoframeworks²⁸, Au clusters through film deposition²⁹, and film annealed-induced Au islands³¹.”

“The branching fractal model is exhibited in Fig. 1d. A single PUF tag is composed of distinct fractal objects, and several iterated bifurcations make up one fractal. Upon each iteration, two new branches are added to each terminal branch. The branch orientations and lengths of the fractals are randomly varied and the symmetry is broken, but exhibiting statistical feature similarity in different bifurcations, as shown in one quasi-Y shape iterated network in Fig. 1d.”

C3: Going to the calculation of the encoding capacity of 10630 (line 243 of the paper) this comes from the assumption that two different “PUFs” can differentiate by one pixel (out of 10 6) and one grayscale tone (out of 255). Obviously, this number is totally unrealistic taking into account the

actual properties of the imaging unit (optical interrogator) in terms of modulation transfer function of the optics or the noise of the detection unit, etc. Moreover such minor differences would result in very high cross correlation between the corresponding patterns, reducing radically the number of usable uncorrelated tags.

R3: Thank the review for pointing out this concern. However, signal interference from the imaging unit is a common issue for nearly all kinds of image-based PUF labels, which has been gradually improved through the optimization of current imaging technologies. The signal interference (decrease of image pixel number or value, gaussian noise, blurring, etc.) could be caused by the instability of the CCD or CMOS sensor in the imaging unit, which can't be completely avoided. But here we would like to respectively illustrate the coping strategy from the label itself and authentication system design.

Firstly, we realize that the image sizes of 1000×1000 can't properly match with the actual response image. Therefore, we rationally decrease the sizes to 750×750 with an encoding capacity of 10^{353} (the basic encoding capacity shall be 10^{20} or larger for PUF system). It's worth noting that the encoding capacity of 10^{353} is indeed an ideal maximum without the consideration of the imaging interference.

The image pixel number, or the image resolution is the key factor in encoding capacity calculation. With a decreased image resolution caused by the external interference, an encoding capacity can be decreased if the physical object size exceeds pattern resolution (*Nat. Rev. Chem.* 1, 0031, 2017). First and foremost, the tag must be physically robust and the feature information must be sufficiently dense so that each PUF key is truly unique and immune to the decrease of resolution caused by the imaging unit. Firstly, our Au-based PUF tag is physically robust against extreme condition, such as high temperature, mechanical abrasion, and contamination. More importantly, we can flexibly improve the encoding capacity by increasing the Au network density in a single PUF, as shown in Fig. 2a-d and 3c in the manuscript. The unlimited density of the physical feature can meet the challenge of the loss of pixels.

Besides, the image noise and contrast can influence the feature extraction of our PUFs, which can finally influence the pattern filling ratio in encoding capacity calculation. In terms of the authentication system, those conditions can be implemented via image processing methods before training and validating. Specifically, deep learning methods have the capability of automatic denoising and deblurring, like the Nonlinear Activation Free Network (NAFNet) concept proposed by Sun, J. et al. For the end-user, too extreme influence from those conditions can scarcely occur, as our system only generates authentication result until a desirable image is acquired. A prompt asking for the end-user to keep the phone steady may be necessary. In our authentication system, if the uploaded image is invalidated, the client is reminded by the software to capture the original PUF again with a better imaging quality for further identification. Importantly, we have improved the AI training model to cover the influence of gaussian noise and further blurring through image preprocessing, the corresponding test results are shown in **Comment 4**. Besides, in order to avoid the loss of grayscale value (0-255) and the decrease of image contrast caused by the imaging unit, we preprocessed the PUF image via grayscale stretch before the AI model training (shown in Figure R9), so that we can fully utilize the grayscale values from 0 to 255 in the encoding capacity calculation. This can also cover the influence of different brightness and contrast of the images in the following PUF authentication.

Moreover, the feature information in one PUF is extremely dense due to the indeterministic

production process. Feature similarity algorithm (FSIM) is based on feature point matching and very sensitive to the image pixel distribution, brightness, contrast, rotation, etc. Under the basic condition of the image resolution that makes the network structure distinguishable, it's almost impossible to generate a high cross correlation between two entirely different PUFs by FSIM, although one captured image may have local pixel distortion or loss due to the poor imaging quality. This is because the PUF pattern has dense pixel distribution and rich grayscale information. Besides, the mass production trait of the PUFs can ensure the tolerance of the poor image quality and keep a high yield.

“Noting that the captured image size and contrast may be decreased by inevitable interference from imaging unit, which can be alleviated by employing PUFs with denser feature information or image preprocessing of denoising and grayscale stretch.”

Figure R9. The grayscale distribution before **a** and after **b** the image preprocessing of grayscale stretch.

C4: Concerning the authentication method, as the authors mention, a conventional image processing analysis could be used to extract the image features of the tag. The authors have developed a modified deep learning scheme for this purpose. Here also the paper suffers from the negligible number of samples. For example, for the generation of the training set they used fifteen out of the twenty sample patterns and they enriched them with additional patterns coming from rotation of the initial 15 patterns by one degree. I'm not sure if this is the best training approach but again a deep learning system based on a total of 20 samples is out of the question. Moreover, I'm wondering what are the benefits of the ML method compared to the conventional image processing methods.

R4: We agree with the reviewer about this concern. In order to provide a convincing model, we changed two important parts of our work:

1) The architecture of the model is changed to a ResNet50 based classification model pretrained by ImageNet (3.2 million images). The input image is preprocessed in grayscale by modification of the pixel distribution and sent to a ResNet50 based classification model. Some random Gaussian noise was added to the training image in order to limit overfitting and also for

covering the influence from the image noise. To reduce the training time and guarantee the generalizability of the model, the pretrained model parameters on ImageNet is used to initialize the ResNet50 and a base model is trained via 1100 PUFs. With a new PUF, only one parameter is added to the last classification layer of the base model, and only this final layer is updated during the training of the new PUF. Noting that both the original PUFs in the database and newly added PUFs are used to update the model to avoid the model forgetting the previous database. The update of only the last layer ensures a fast training procedure. When the fifth new image was added, the training can be finished in 5.48 s (total 40 epochs, 0.137s for each epoch, Figure R5b) in the database expansion test. The results of the database expansion test are shown below.

2) A lot more PUF images are captured (1850 PUFs): 1) **1300** PUFs for the establishment of the database with 1100 PUFs for building the base database and 200 PUFs for the expansion of the database; 2) **550** PUFs as the fake PUFs out of the database and augmented for training a general model. The TEST_Set contains **37000** images augmented from the original 1850 PUFs, which considers different conditions (brightness, rotations, and noise) that can occur in practice, covering the influence from different capturing conditions and imaging unit. We truly know that not all conditions can be considered in this test, but our work aims to provide a benchmark to prove that our model and technique have the potential application in the PUF authentication.

The corresponding training/validation/testing results are shown below. According to the results, we can achieve a 0% wrong validation ratio of the genuine images (0% false positives and 5% false positives). Noting that the high image invalidation rate comes from the large ratio of the fake image (30%), which is tested as the invalidation in our model setup. Of course, the further optimization of the model for obtaining a higher accuracy is being implemented. Besides, we have made a list of the different database sizes for AI model training in other PUF works.

Two added references related to the modified model are as follows,

[1] ImageNet:

J. Deng, W. Dong, R. Socher, L. -J. Li, Kai Li and Li Fei-Fei, "ImageNet: A large-scale hierarchical image database," 2009 IEEE Conference on Computer Vision and Pattern Recognition, 2009, pp. 248-255, doi: 10.1109/CVPR.2009.5206848.

[2] ResNet50:

K. He, X. Zhang, S. Ren and J. Sun, "Deep Residual Learning for Image Recognition," 2016 IEEE Conference on Computer Vision and Pattern Recognition (CVPR), 2016, pp. 770-778, doi: 10.1109/CVPR.2016.90.

Table 1. The parameters for the AI model training/validation/testing.

Dataset	Size	Augmentation	Labels
Original images (ORI_Set)	1850 PUFs		
Training set (TRAIN_Set)	1100 PUFs from ORI_Set	Rotation: 0, 30, 60, 90, .. , 330°	Corresponding classes: 0-1099
Validation set (VAL_Set)	The same PUFs with TRAIN_Set	Rotation: 1, 3, 5, 7, .. , 359°	Corresponding classes: 0-1099
Candidate set (CAND_Set)	200 PUFs from ORI_Set	Rotation: 0, 30, 60, 90, .. , 330° for training and 1, 3, 5, 7, .. , 359° for validation	Corresponding classes: 1100-1299
Fake set (FAKE_Set)	550 PUFs from ORI_Set	1) Rotation: 10 angles are randomly selected from those that are not considered in the TRAIN_Set and VAL_Set. 2) Gaussian noise for each image 3) 10 different brightness conditions applied to each image	All labelled as "FAKE"
Test set (TEST_Set)	1850 PUFs of ORI_Set with augmentations that generate 37000 images		Corresponding classes: 0-1299 for the first 1300 PUFs in the database, and "FAKE" for the left 550 PUFs

Figure R5. **a** The training procedures of the base classification model (TRAIN_Set and VAL_Set). **b** The training time of the classification model as a function of the newly added 200 PUFs, showing a gradually stable training epoch below 40. The inset is the plot of training and validation accuracy of the classification model with the first newly added image. The training stops when the validation accuracy reaches at the threshold of 95%. R_c represents the accuracy. **c** Correct/Wrong validation ratio and inactivation ratio of the 37000 images (TEST_Set) as a function of similarity threshold. The ideal maximum of the correct accuracy is 0.7, as the whole test dataset contains 30% fake images. The 0% wrong validation can be achieved beyond the threshold of 0.5.

Table 2 The database size for AI model training in different works of PUFs.

PUF type	Database size for AI model training	Reference
Crumpling of 2D Materials	500	Matter 3, 2160–2180, 2020.
Drop-casting nanoparticle tags	1020	ACS Nano 15, 2901–2910, 2021.
Laser-induced polymer wrinkle	112	Adv. Mater. 32, 2003032, 2020.
Multi-functional nanoinks	100	Nano Today 41, 101324, 2021.
Random wrinkles	6	ACS Appl. Mater. Interfaces 13, 27548–27556, 2021.
Inkjet-printed unclonable quantum dot	6	Nat. Commun. 10, 2409, 2019.
Inkjet-printed unclonable quantum dot	10	ACS Appl. Mater. Interfaces 13, 15701–15708, 2021.
Unclonable photonic crystal hydrogels	10	ACS Appl. Mater. Interfaces 14, 2369–2380, 2022.
Our work	1300	

The further optimization of the model (for getting a higher accuracy) and testing of more images with different conditions are underway (images with different scaling and blurring are to be tested). As for the database expansion test, more new PUFs (another 300 images) are to be supplemented. This part of the works is expected to be finished in two weeks.

The deep learning method has many advantages compared to the conventional image processing methods, as follows,

1) the deep neural networks can record the information of unclonable features so that the pattern verification can be completed only through the neural network, considerably improving the speed and accuracy. On the contrary, the conventional image processing method (e.g. the similarity methods) is time-consuming especially when the database is huge as all the images in the database need to be compared with the input image from the user during the authentication.

2) deep learning, as a black box with a few explanations of the functioning mechanism, is an advantage for unclonable anti-counterfeiting technique because it is a tamper proof.

3) as for the conventional image processing, a rectangular coordinate system is required to define the position and orientation of each physical feature, e.g. the input image usually needs to be rotated in different angles to search the matching PUF in the database. However, any rotation of the security label results in a completely different code (Fig. S28; *Sci. Adv.* 4, e1701384, 2018). In other word, a real security label will be recognized as a fake one if the end-user rotates or zooms in/out it

during the pattern readout process. Alternatively, a mark on the security label is used to define xy axis, which will rotate as the security label rotates (*Adv. Mater.* 28, 2330–2336, 2016). This guarantees only one code for each security label. By contrast, the deep learning-based authentication system allows the end-user to readout the patterns with different image brightness, amplifications and rotations, as AI can generate the scale and rotation invariant features of the security labels. Although, it is time-consuming to train the deep learning machine to learn the characteristic features of the security labels and the repeated model training is required for each new PUF pattern. In our work, mass of new images are further added to the deep learning model successively and a gradually shortened training time is demonstrated, allowing for a potentially unlimited new image amount with a low time cost. This strategy helps to break the trade-off between the large key database of the PUF system and long training time of the deep learning model in practice.

“Figure 4a schematically demonstrates the deep learning-based authentication system of the security label. Firstly, each PUF pattern is captured by the manufacturers by using the microscope. The different images are then preprocessed by modification of the pixel distribution and devoted to a ResNet50 based classification model⁴⁷ for learning the characteristics of the PUF patterns. To reduce the training time and guarantee the generalizability of the model, the pretrained model parameters on ImageNet⁴⁸ is used to initialize the ResNet50. Next, the images are classified in a general manner and stored in the database for subsequent authentication.”

“For experimentally demonstrating the above authentication system, 1300 different PUF tags were randomly captured to establish the security key database (parts of PUFs shown in Supplementary Fig. 22). 1100 PUFs of them were used for the establishment of the basic database, and the rest 200 PUFs were used for the key expansion test of the database. Each of the 1100 PUFs were rotated with different angles, forming 211200 images as the dataset for deep learning model training (13200 images with less rotations for training and 198000 images for validation). Every input image is preprocessed by grayscale stretch and added random noise to avoid overfitting.”

“For testing our authentication system, the above 1300 PUF tags (in the database) in different conditions (brightness, rotation angle, and random noise, total of 26000 images, Supplementary Fig. 24) and 11000 images from 550 new PUF tags (not in the database) were captured and uploaded to the trained AI for a test. We investigated the validation ratio as a function of the similarity threshold by testing multiple similarity indexes for obtaining the best threshold of validation. Figure 4c reveals that when the similarity threshold reaches 0.5, we can achieve a 0% wrong validation ratio of the genuine images (that is, the rate of false positives is 0%) and 5% false negatives. In this work, we set the similarity threshold at a value of 0.5, and the authentication of a PUF tag with an encoding capacity of about 10^{353} can be finished in 6.36 seconds.”

“The conventional image processing algorithms are based on the matching of feature points in library files one by one^{1,37}. Apart from the relatively tedious matching time, their performance also strongly depends on the image orientation and quality^{36,38}. Deep learning (DL)^{11,13,17}, as an artificial intelligence (AI) technique, has been popularly used to validate the security key through the trained neural networks with high authentication efficiency and accuracy as well as high readout toleration in different conditions.”

REVIEWER COMMENTS

Reviewer #1 (Remarks to the Author):

The authors have submitted a revised version of their manuscript entitled "Random fractal-enabled physical unclonable functions with dynamic AI authentication". The authors have made numerous revisions in response to a prior review.

The first innovation of this work stems from the development of a fractal based graphical PUF with large grayscale encoding capacity. This is contrasted to wrinkling based and luminescent-type graphical PUFs in the recent literature which are argued to exhibit poor robustness and/or durability.

The second innovation of this work is the implementation of a dynamic database strategy which is developed and employed to train a deep-learning authentication system. This strategy avoids the need to re-train with the entire database when new images or graphical PUF data are generated. Hence, new devices/products can be continuously added to the database with limited training time/cost.

Thirdly a Raman mapping is shown as a proof-of-concept pathway to potentially adding additional information/complexity to the graphical PUF by adding data that is in principal orthogonal to the PUF image (relying on nanoscale roughness).

Regarding prior Review comments:

The authors have adequately addressed the majority of my prior concerns/comments. Including those related to the potential Raman aspect, convincingness of the AI authentication protocol etc. However, one critical item, regarding the reliance/usage of grayscale information has only been partially suitably addressed in response to Reviewer 1 Comments 1 and 2. The remaining concern is regarding the heavy reliance on 0-255 levels in computing the encoding capacity, as echoed by Reviewer 2 and detailed below.

Comments/concerns:

(1) Reviewer 2 also previously raised an excellent point regarding encoding capacity (comment C3). The reported encoding capacity indeed appears somewhat optimistic. The authors have made some acknowledgments and discussion of the practical nature of this point, but it is not clear if they have made satisfactory edits to the main text. According to reference #1 (Carro-Temboury et al) of the supporting information, the encoding capacity is generally less than the upper theoretical limit based on the image capacity ($\# \text{ pixel values}^{\# \text{ number of pixels}}$) and a master equation is provided to estimate the practical encoding capacity. In the supporting information Supplementary Note #1 the authors utilize the Carro-Temboury method/equation to estimate their encoding capacity. However, Carro-Temboury utilized $C = 2, 4, \text{ or } 7$ based on their particular PUF. Meanwhile the present work assumes $C = 255$. Per reviewer 2's comment, it is not clear if using $C = 255$ represents a fair estimate of the color resolution and hence the encoding capacity, especially when considering practical intensity resolution factors and the pre/post-processing histograms of Supplementary Fig 8. Perhaps the authors could either: (1) better justify the use of this $C = 255$ value and reiterate its assumption in the calculation in the main text (near key lines 224-226 of page 9), or (2) could revise the calculation using a more conservative estimate for C , perhaps based on a conservative estimate of the practical grayscale resolution.

New comments:

(NC1) Upon close inspection, the literature introduction and citations need improvement.

"Electronic PUF" misses a whole body of literature, see relevant review articles that can be cited, for

example:

(A) Gao, Y., Al-Sarawi, S.F. & Abbott, D. Physical unclonable functions. Nat Electron 3, 81–91 (2020).

(B) C. -H. Chang, Y. Zheng and L. Zhang, "A Retrospective and a Look Forward: Fifteen Years of Physical Unclonable Function Advancement," in IEEE Circuits and Systems Magazine, vol. 17, no. 3, pp. 32-62, thirdquarter 2017, doi: 10.1109/MCAS.2017.2713305.

"Spectral PUF" misses some recent photonics-based examples in the literature that could be cited, e.g.:

(A) Brian C. Grubel, Bryan T. Bosworth, Michael R. Kossey, Hongcheng Sun, A. Brinton Cooper, Mark A. Foster, and Amy C. Foster, "Silicon photonic physical unclonable function," Opt. Express 25, 12710-12721 (2017)

(B) Tarik, F.B., Famili, A., Lao, Y. et al. Scalable and CMOS compatible silicon photonic physical unclonable functions for supply chain assurance. Sci Rep 12, 15653 (2022).

(C) "Harnessing disorder for photonic device applications" [Appl. Phys. Rev. 9, 011309 (2022)]

(NC2) English/grammar needs improvement in various areas, below are some examples needing improvement along side possible suggestion or interpretation in parenthesis:

Page 1, line 17 - "often causes it difficult" (often makes it difficult)

Page 1, line 24 - "can be served as the" (can serve as)

Page 2, line 30 - "Anti-fake labels as the certification of products retain an increasing challenge in authenticity" (unclear)

Page 6, line 167-169 (unclear)

Page 15, line 367 - "Raman characterization is potential" (has potential)^[1]_{ISEP}

^[1]_{ISEP}

Reviewer #2 (Remarks to the Author):

The authors reacted in a satisfying way to the concerns and comment of this reviewer and I think the paper can be accepted for publication to the Nature Communications journal

Response Letter

We are grateful for the positive comments and helpful suggestions from the reviewers. Our responses to the reviewers' comments and the changes made in our manuscript are as follows. For the sake of readability, our responses are highlighted by blue font and the changes to our manuscript are presented in green.

Reviewer #1 (Remarks to the Author):

The authors have submitted a revised version of their manuscript entitled "Random fractal-enabled physical unclonable functions with dynamic AI authentication". The authors have made numerous revisions in response to a prior review.

The first innovation of this work stems from the development of a fractal based graphical PUF with large grayscale encoding capacity. This is contrasted to wrinkling based and luminescent-type graphical PUFs in the recent literature which are argued to exhibit poor robustness and/or durability.

The second innovation of this work is the implementation of a dynamic database strategy which is developed and employed to train a deep-learning authentication system. This strategy avoids the need to re-train with the entire database when new images or graphical PUF data are generated. Hence, new devices/products can be continuously added to the database with limited training time/cost.

Thirdly a Raman mapping is shown as a proof-of-concept pathway to potentially adding additional information/complexity to the graphical PUF by adding data that is in principal orthogonal to the PUF image (relying on nanoscale roughness).

Regarding prior Review comments:

The authors have adequately addressed the majority of my prior concerns/comments. Including those related to the potential Raman aspect, convincingness of the AI authentication protocol etc. However, one critical item, regarding the reliance/usage of grayscale information has only been partially suitably addressed in response to Reviewer 1 Comments 1 and 2. The remaining concern is regarding the heavy reliance on 0-255 levels in computing the encoding capacity, as echoed by Reviewer 2 and detailed below.

We truly appreciate the reviewer for the enthusiastic support and precious comments of our manuscript. We have carefully revised the manuscript based on the constructive suggestions.

C1: Reviewer 2 also previously raised an excellent point regarding encoding capacity (comment C3). The reported encoding capacity indeed appears somewhat optimistic. The authors have made some acknowledgments and discussion of the practical nature of this point, but it is not clear if they have made satisfactory edits to the main text. According to reference #1 (Carro-Temboury et al) of the supporting information, the encoding capacity is generally less than the upper theoretical limit

based on the image capacity ($\# \text{ pixel values}^{\# \text{ number of pixels}}$) and a master equation is provided to estimate the practical encoding capacity. In the supporting information Supplementary Note #1 the authors utilize the Carro-Temboury method/equation to estimate their encoding capacity. However, Carro-Temboury utilized $C = 2, 4, \text{ or } 7$ based on their particular PUF. Meanwhile the present work assumes $C = 255$. Per reviewer 2's comment, it is not clear if using $C = 255$ represents a fair estimate of the color resolution and hence the encoding capacity, especially when considering practical intensity resolution factors and the pre/post-processing histograms of Supplementary Fig 8. Perhaps the authors could either: (1) better justify the use of this $C = 255$ value and reiterate its assumption in the calculation in the main text (near key lines 224-226 of page 9), or (2) could revise the calculation using a more conservative estimate for C , perhaps based on a conservative estimate of the practical grayscale resolution.

R1: We thank the reviewer for proposing this critical issue and the specific suggestions. We really agree with the reviewer that the calculation of the encoding capacity should be based on a more practical grayscale resolution, considering that the practically captured PUF image cannot possess a theoretical grayscale distribution from 0 to 255 due to the limitation of the imaging unit (shortage in image brightness/contrast/color resolution). Particularly, the grayscale level of 255 can be achieved if the imaging unit provides a desirable contrast between the feature structure and background (substrate), which puts forward a high requirement for the imaging device and image quality.

Here we followed the **second suggestions** from the reviewer. Based on the practical grayscale distribution of the captured PUF image calculated by the histograms in Supplementary Fig 8, we tried to define a conservative level of grayscale intensity in one pixel as the final "C" in the encoding capacity calculation. In order to obtain a more practical "C", we randomly selected 10 PUF images from the basic database (all of them were captured under normal imaging conditions) and calculated their grayscale distributions, as shown in Figure R1. We defined the peak of grayscale histogram as the maximum of background grayscale, which means that the background loads the largest number of pixels with consistent grayscale intensity and the above grayscale intensities are all from the PUF feature structures. After subtracting the invalid grayscale information from the background and the high grayscale intensities that the practical images cannot reach, we defined the grayscale range of the feature structures from 74 (average maximum of background grayscale) to 214 (average maximum of feature grayscale), and the value of "C" is set as the difference, i.e., 140. The corresponding encoding capacity calculations have been revised in the manuscript (Figure 3c) and Supplementary Information (Supplementary Note 1). For example, after the correcting of "C", the encoding capacity of PUF tag in Supplementary Fig 9 has changed from 10^{353} to 10^{348} , showing a little decrease in capacity.

Figure R1. Calculation of practical grayscale intensity level of the original PUF image. a Grayscale image of a typical PUF pattern. **b** Grayscale distribution histogram of a typical PUF pattern. **c** Grayscale distributions from 10 randomly selected PUF images in the basic database. **d** Maximums of feature grayscale and background grayscale from the 10 PUF images in **c**.

“However, due to the inevitable limitation of the imaging unit in image contrast or color resolution, the physical features in practically captured PUF images usually cannot possess a theoretical grayscale distribution from 0 to 255. To present a more general level of grayscale encoded in physical features, we calculated the grayscale histograms from 10 randomly selected PUF images in the basic database, subtracted the grayscale information that the feature structure cannot reach, and counted an average value of 140 as the valid level of grayscale in a practical PUF image (valid grayscale distribution from 74 to 214, as shown in Supplementary Fig. 8).”

C2: Upon close inspection, the literature introduction and citations need improvement.

“Electronic PUF” misses a whole body of literature, see relevant review articles that can be cited, for example:

(A) Gao, Y., Al-Sarawi, S.F. & Abbott, D. Physical unclonable functions. *Nat Electron* 3, 81–91 (2020).

(B) C. -H. Chang, Y. Zheng and L. Zhang, "A Retrospective and a Look Forward: Fifteen Years of Physical Unclonable Function Advancement," in *IEEE Circuits and Systems Magazine*, vol. 17, no. 3, pp. 32-62, thirdquarter 2017, doi: 10.1109/MCAS.2017.2713305.

“Spectral PUF” misses some recent photonics-based examples in the literature that could be cited, e.g.:

(A) Brian C. Grubel, Bryan T. Bosworth, Michael R. Kossey, Hongcheng Sun, A. Brinton Cooper,

Mark A. Foster, and Amy C. Foster, "Silicon photonic physical unclonable function," *Opt. Express* **25**, 12710-12721 (2017)

(B) Tarik, F.B., Famili, A., Lao, Y. et al. Scalable and CMOS compatible silicon photonic physical unclonable functions for supply chain assurance. *Sci Rep* **12**, 15653 (2022).

(C) "Harnessing disorder for photonic device applications" [*Appl. Phys. Rev.* **9**, 011309 (2022)]

R2: We thank the review for recommending the relevant references about electronic PUFs and silicon photonic PUFs. As suggested, we have made a careful research on the development of electronic and silicon photonic PUFs and included the relevant descriptions and references into the Introduction in the manuscript.

"(ii) spectral PUFs...irregular texture²⁵/matrix^{5,26} linear scattering-based speckle patterns, and chaotic nonlinear silicon photonic devices^{27,28}); and (iii) complex electronic PUFs with diverse disorders and inherent imperfections (e.g., graphene²⁹ or randomly distributed carbon nanotube-based³⁰ field-effect transistors, oxide or halide-based memristors with intrinsic entropy sources^{6,31,32})."

25. Kim, M. S. et al. Revisiting silk: a lens-free optical physical unclonable function. *Nat. Commun.* **13**, 247 (2022).

26. Cao, H. & Eliezer, Y. Harnessing disorder for photonic device applications. *Appl. Phys. Rev.* **9**, 011309 (2022).

27. Grubel, B. C. et al. Silicon photonic physical unclonable function. *Opt. Express* **25**, 12710 (2017).

28. Tarik, F. B. et al. Scalable and CMOS compatible silicon photonic physical unclonable functions for supply chain assurance. *Sci. Rep.* **12**, 15653 (2022).

29. Dodda, A. et al. Graphene-based physically unclonable functions that are reconfigurable and resilient to machine learning attacks. *Nat. Electron.* **4**, 364–374 (2021).

30. Zhong, D. et al. Twin physically unclonable functions based on aligned carbon nanotube arrays. *Nat. Electron.* **5**, 424–432 (2022).

31. Chang, C., Zheng, Y. & Zhang, L. A retrospective and a look forward: Fifteen years of physical unclonable function advancement. *IEEE Circuits Syst. Mag.* **17**, 32–62 (2017).

32. John, R. A. et al. Halide perovskite memristors as flexible and reconfigurable physical unclonable functions. *Nat. Commun.* **12**, 3681 (2021).

C3: English/grammar needs improvement in various areas, below are some examples needing improvement along side possible suggestion or interpretation in parenthesis:

Page 1, line 17 - "often causes it difficult" (often makes it difficult)

Page 1, line 24 - "can be served as the" (can serve as)

Page 2, line 30 - "Anti-fake labels as the certification of products retain an increasing challenge in authenticity" (unclear)

Page 6, line 167-169 (unclear)

Page 15, line 367 - "Raman characterization is potential" (has potential)

R3: We thank the reviewer for the thoughtful comments. As suggested, we have revised the whole manuscript and carefully proof-read the manuscript to improve English and clarity. We also sent the manuscript to a language polishing service for English language editing during the revision process. The corresponding Editing Certificate has uploaded as a supplementary file for editor and reviewer only. We believe that the language is now acceptable for the review process and publication. All the revisions in grammar/English have been highlighted in green in the Revised and marked manuscript, and the revision will not change the original meanings of the text.

Two proposed examples:

“Anti-fake labels as the certification of products retain an increasing challenge in authenticity” has changed to

“Anti-fake labels as the authentication tools of product authenticity face an increasing challenge in security”

“The branch orientations and lengths of the fractals are randomly varied and the symmetry is broken, but exhibiting statistical feature similarity in different bifurcations, as shown in one quasi “Y-shape” iterated network in Fig. 1d.” has changed to

“The extension directions and lengths of the branches are randomly varied but exhibit statistical feature similarity in different bifurcations. For example, in Fig. 1d, the bifurcations of part 1 and the derived part 2 have the similar quasi “Y-shape” feature.”

Reviewer #2 (Remarks to the Author):

The authors reacted in a satisfying way to the concerns and comment of this reviewer and I think the paper can be accepted for publication to the Nature Communications journal.

Thanks so much for the reviewer’s recognition and putting so many efforts in reviewing our manuscript.

REVIEWERS' COMMENTS

Reviewer #1 (Remarks to the Author):

The authors have suitably addressed my remaining concerns and I now recommend this work for timely publication.

Reviewer #1:

The authors have suitably addressed my remaining concerns and I now recommend this work for timely publication.

Our response:

Thanks so much for the reviewer's recognition and putting so many efforts in reviewing our manuscript.